# A Rigorous Link between Deep Ensembles and (Variational) Bayesian Methods

**Veit D. Wild**[*]
University of Oxford

**Sahra Ghalebikesabi**
University of Oxford

**Dino Sejdinovic**
University of Adelaide

**Jeremias Knoblauch**
University College London

## Abstract

We establish the first mathematically rigorous link between Bayesian, variational Bayesian, and ensemble methods. A key step towards this is to reformulate the non-convex optimisation problem typically encountered in deep learning as a convex optimisation in the space of probability measures. On a technical level, our contribution amounts to studying generalised variational inference through the lens of Wasserstein gradient flows. The result is a unified theory of various seemingly disconnected approaches that are commonly used for uncertainty quantification in deep learning—including deep ensembles and (variational) Bayesian methods. This offers a fresh perspective on the reasons behind the success of deep ensembles over procedures based on standard variational inference, and allows the derivation of new ensembling schemes with convergence guarantees. We showcase this by proposing a family of interacting deep ensembles with direct parallels to the interactions of particle systems in thermodynamics, and use our theory to prove the convergence of these algorithms to a well-defined global minimiser on the space of probability measures.

## 1 Introduction

A major challenge in modern deep learning is the accurate quantification of uncertainty. To develop trustworthy AI systems, it will be crucial for them to recognize their own limitations and to convey the inherent uncertainty in the predictions. Many different approaches have been suggested for this. In variational inference (VI), a prior distribution for the weights and biases in the neural network is assigned and the best approximation to the Bayes posterior is selected from a class of parameterised distributions (Graves, 2011; Blundell et al., 2015; Gal and Ghahramani, 2016; Louizos and Welling, 2017). An alternative approach is to (approximately) generate samples from the Bayes posterior via Monte Carlo methods (Welling and Teh, 2011; Neal, 2012). Deep ensembles are another approach, and rely on a train-and-repeat heuristic to quantify uncertainty (Lakshminarayanan et al., 2017).

Much ink has been spilled over whether one can see deep ensembles as a Bayesian procedure (Wilson, 2020; Izmailov et al., 2021; D'Angelo and Fortuin, 2021) and over how these seemingly different methods might relate to each other. Building on this discussion, we shed further light on the connections between Bayesian inference and deep ensemble techniques by taking a different vantage point. In particular, we show that methods as different as variational inference, Langevin sampling (Ermak, 1975), and deep ensembles can be derived from a well-studied generally infinite-dimensional regularised optimisation problem over the space of probability measures (see e.g. Guedj and Shawe-Taylor, 2019; Knoblauch et al., 2022). As a result, we find that the differences between

37th Conference on Neural Information Processing Systems (NeurIPS 2023).

---
[*]veit.wild@stats.ox.ac.uk

these algorithms map directly onto different choices regarding this optimisation problem. Key differences between the algorithms boil down to different choices of regularisers, and whether they implement a finite-dimensional or infinite-dimensional gradient descent. Here, finite-dimensional gradient descent corresponds to parameterised VI schemes, whilst the infinite-dimensional case maps onto ensemble methods.

The contribution of this paper is a new theory that generates insights into existing algorithms and provides links between them that are mathematically rigorous, unexpected, and useful. On a technical level, our innovation consists in analysing them as algorithms that target an optimisation problem in the space of probability measures through the use of a powerful technical device: the Wasserstein gradient flow (see e.g. Ambrosio et al., 2005). While the theory is this paper's main concern, its potentially substantial methodological payoff is demonstrated through the derivation of a new inference algorithm based on gradient descent in infinite dimensions and regularisation with the maximum mean discrepancy. We use our theory to show that this algorithm—unlike standard deep ensembles—is derived from a strictly convex objective defined over the space of probability measures. Thus, it targets a unique minimum, and is capable of producing samples from this global minimiser in the infinite particle and time horizon limit. This makes the algorithm provably convergent; and we hope that it can help plant the seeds for renewed innovations in theory-inspired algorithms for (Bayesian) deep learning.

The paper proceeds as follows: Section 2 discusses the advantages of lifting losses defined on Euclidean spaces into the space of probability measures through a generalised variational objective. Section 3 introduces the notion of Wasserstein gradient flows, while Section 4 links them to the aforementioned objective and explains how they can be used to construct algorithms that bridge Bayesian and ensemble methods. The paper concludes with Section 5, where the findings of the paper are illustrated numerically.

## 2 Convexification through probabilistic lifting

One of the most technically challenging aspects of contemporary machine learning theory is that the losses $\ell : \mathbb{R}^J \to \mathbb{R}$ we wish to minimise are often highly non-convex. For instance, one could wish to minimise $\ell(\theta) := \frac{1}{N} \sum_{n=1}^{N} \left( y_n - f_\theta(x_n) \right)^2$ where $\{(x_n, y_n)\}_{n=1}^{N}$ is a set of paired observations and $f_\theta$ a neural network with parameters $\theta$. While deep learning has shown that non-convexity is often a negligible *practical* concern, it makes it near-impossible to prove many basic *theoretical* results that a good learning theory is concerned with, as $\ell$ has many local (or global) minima (Fort et al., 2019; Wilson and Izmailov, 2020). We reintroduce convexity by lifting the problem onto a computationally more challenging space. In this sense, the price we pay for the convenience of convexity is the transformation of a finite-dimensional problem into an infinite-dimensional one, which is numerically more difficult to tackle. Figure 1 illustrates our approach:

$$\min_{\theta \in \Theta} \ell(\theta) \qquad \min_{Q \in \mathcal{P}(\mathbb{R}^J)} \int \ell(\theta) dQ(\theta) \qquad \min_{Q \in \mathcal{P}(\mathbb{R}^J)} \left\{ \int \ell(\theta) dQ(\theta) + \lambda D(Q, P) \right\}$$

**Step 1:** probabilistic lifting     **Step 2:** convexification through regularisation

Figure 1: Illustration of convexification through probabilistic lifting.

First, we transform a non-convex optimisation $\min_{\theta \in \Theta} \ell(\theta)$ into an infinite-dimensional optimisation over the set of probability measures $\mathcal{P}(\mathbb{R}^J)$, yielding $\min_{Q \in \mathcal{P}(\mathbb{R}^J)} \int \ell(\theta) dQ(\theta)$. As an integral, this objective is linear in $Q$. However, linear functions are not strictly convex. We therefore need to add a strictly convex regulariser to ensure uniqueness of the minimiser.[2] We prove in Appendix A that indeed—for a regulariser $D : \mathcal{P}(\mathbb{R}^J) \times \mathcal{P}(\mathbb{R}^J) \to [0, \infty]$ such that $(Q, P) \mapsto D(Q, P)$ is strictly convex and $P \in \mathcal{P}(\mathbb{R}^J)$ a fixed measure—existence and uniqueness of a global minimiser can be guaranteed. Given a scaling constant $\lambda > 0$, we can now put everything together to obtain

---

[2]To illustrate why this is necessary, assume that there are $\theta_A$ and $\theta_B$ in $\mathbb{R}^J$ so that $\ell(\theta_A) = \ell(\theta_B) = \min_{\theta \in \Theta} \ell(\theta)$. Due to linearity, each measure $Q_t \in \mathcal{P}(\mathbb{R}^J)$, $t \in [0, 1]$, defined as $Q_t := (1 - t)\delta_{\theta_A} + t\delta_{\theta_B}$ provides one (of the infinitely many) global minima in the set $\arg \min_{Q \in \mathcal{P}(\mathbb{R}^J)} \int \ell(\theta) \, dQ(\theta)$.

the loss $L$ and the unique minimiser $Q^*$ as

$$L(Q) := \int \ell(\theta)\, dQ(\theta) + \lambda D(Q, P), \qquad Q^* := \underset{Q \in \mathcal{P}(\mathbb{R}^J)}{\arg\min}\, L(Q). \tag{1}$$

Throughout, whenever $Q$ and $Q^*$ have an associated Lebesgue density, we write them as $q$ and $q^*$. Moreover, all measures, densities, and integrals will be defined on the parameter space $\mathbb{R}^J$ of $\theta$. Similarly, the gradient operator $\nabla$ will exclusively denote differentiation with respect to $\theta \in \mathbb{R}^J$.

## 2.1 One objective with many interpretations

In the current paper, our sole focus lies on resolving the difficulties associated with non-convex optimisation of $\ell$ on Euclidean spaces. Through probabilistic lifting and convexification, we can identify a unique minimiser $Q^*$ in the new space, which minimises the $\theta$-averaged loss $\theta \mapsto \ell(\theta)$ without deviating too drastically from some reference measure $P$. In this sense, $Q^*$ summarises the quality of all (local and global) minimisers of $\ell(\theta)$ by assigning them a corresponding weight. The choices for $\lambda$, $D$ and $P$ determine the trade-off between the initial loss $\ell$ and reference measure $P$ and therefore the weights we assign to different solutions.

Yet, (1) is not a new problem form: it has various interpretations, depending on the choices for $D, \lambda, \ell$ and the framework of analysis (see e.g. Knoblauch et al., 2022, for a discussion). For example, if $\ell(\theta)$ is a negative log likelihood, $D$ is the Kullback-Leibler divergence (KL), and $\lambda = 1$, $Q^*$ is the **standard Bayesian** posterior, and $P$ is the Bayesian prior. This interpretation of $P$ as a prior carries over to **generalised Bayesian** methods, in which we can choose $\ell(\theta)$ to be any loss, $D$ to be any divergence on $\mathcal{P}(\mathbb{R}^J)$, and $\lambda$ to regulate how fast we learn from data (see e.g. Bissiri et al., 2016; Jewson et al., 2018; Knoblauch et al., 2018; Miller and Dunson, 2019; Knoblauch, 2019; Alquier, 2021a; Husain and Knoblauch, 2022; Matsubara et al., 2022; Wild et al., 2022; Wu and Martin, 2023; Altamirano et al., 2023). In essence, the core justification for these generalisations is that the very assumptions justifying application of Bayes' Rule are violated in modern machine learning. In practical terms, this results in a view of Bayes' posteriors as one—of many possible—measure-valued estimators $Q^*$ of the form in (1). Once this vantage point is taken, it is not clear why one *should* be limited to using only one particular type of loss and regulariser for *every* possible problem. Seeking a parallel with optimisation on Euclidean domains, one may then compare the orthodox Bayesian view with the insistence on only using quadratic regularisation for *any* problem. While it is beyond the scope of this paper to cover these arguments in depth, we refer the interested reader to Knoblauch et al. (2022).

A second line of research featuring objectives as in (1) are **PAC-Bayes** methods, whose aim is to construct generalisation bounds (see e.g. Shawe-Taylor and Williamson, 1997; McAllester, 1999b,a; Grünwald, 2011). Here, $\ell$ is a general loss, but $P$ only has the interpretation of some reference measure that helps us measure the complexity of our hypotheses via $Q \mapsto \lambda D(Q, P)$ (Guedj and Shawe-Taylor, 2019; Alquier, 2021b). Classic PAC-Bayesian bounds set $D$ to be KL, but there has been a recent push for different complexity measures (Alquier and Guedj, 2018; Bégin et al., 2016; Haddouche and Guedj, 2023).

## 2.2 Generalised variational inference (GVI) in finite and infinite dimensions

In line with the terminology coined in Knoblauch et al. (2022), we refer to any algorithm aimed at solving (1) as a **generalised variational inference (GVI)** method. Broadly speaking, there are two ways one could design such algorithms: in finite or infinite dimensions. **Finite-dimensional GVI:** This is the original approach advocated for in Knoblauch et al. (2022): instead of trying to compute $Q^*$, approximate it by solving $Q_{\nu^*} = \arg\min_{Q_\nu \in \mathcal{Q}} L(Q)$ for a set of measures $\mathcal{Q} := \{Q_\nu : \nu \in \Gamma\} \subset \mathcal{P}(\mathbb{R}^J)$ parameterised by a parameter $\nu \in \Gamma \subseteq \mathbb{R}^I$. To find $\mathcal{Q}_\nu$, one now simply performs (finite-dimensional) gradient descent with respect to the function $\nu \mapsto L(Q_\nu)$. For the special case where $P$ is a Bayesian prior, $\lambda = 1$, $\ell(\theta)$ is a negative log likelihood parametrised by $\theta$, and $D = \mathrm{KL}$, this recovers the well-known standard VI algorithm. To the best of our knowledge, all methods that refer to themselves as VI or GVI in the context of deep learning are based on this approach (see e.g. Graves, 2011; Blundell et al., 2015; Louizos and Welling, 2017; Wild et al., 2022). Since procedures of this type solve a finite-dimensional version of (1), we refer to them as **finite-dimensional GVI (FD-GVI)** methods throughout the paper. While such algorithms can perform well, they have some obvious theoretical problems: First of all, the finite-dimensional approach typically forces us to

choose $\mathcal{Q}$ and $P$ to be simple distributions such as Gaussians to ensure that $L(Q_\nu)$ is a tractable function of $\nu$. This often results in a $\mathcal{Q}$ that is unlikely to contain a good approximation to $Q^*$; raising doubt if $Q_{\nu^*}$ can approximate $Q^*$ in any meaningful sense. Secondly, even if $Q \mapsto L(Q)$ is strictly convex on $\mathcal{P}(\mathbb{R}^J)$, the parameterised objective $\nu \in \Gamma \mapsto L(Q_\nu)$ is usually not. Hence, there is no guarantee that gradient descent leads us to $Q_{\nu^*}$. This point also applies to very expressive variational families (Rezende and Mohamed, 2015; Mescheder et al., 2017) which may be sufficiently rich that $Q^* \in \mathcal{Q}$, but whose optimisation problem $\nu \in \Gamma \mapsto L(Q_\nu)$ is typically non-convex and hard to solve, so that no guarantee for finding $Q^*$ can be provided. While this does not necessarily make FD-GVI impractical, it does make it exceedingly difficult to provide a rigorous theoretical analysis outside of narrowly defined settings.

**FD-GVI in function space:** A collection of approaches formulated as infinite-dimensional problems are GVI methods on an infinite-dimensional function space (Ma et al., 2019; Sun et al., 2018; Ma and Hernández-Lobato, 2021; Rodriguez-Santana et al., 2022; Wild et al., 2022). Here, the loss is often convex in function space. In practice however, the variational stochastic process still requires parameterization to be computationally feasible—and in this sense, function space methods are FD-GVI approaches. The resulting objectives require a good approximation of the functional KL-divergence (which is often challenging), and lead to a typically highly non-convex variational optimization problem in the parameterised space.

**Infinite-dimensional GVI:** Instead of minimising the (non-convex) problem $\nu \mapsto L(Q_\nu)$, we want to exploit the convex structure of $Q \mapsto L(Q)$. Of course, a priori it is not even clear how to compute the gradient for a function $Q \mapsto L(Q)$ defined on an infinite-dimensional nonlinear space such as $\mathcal{P}(\mathbb{R}^J)$. However, in the next part of this paper we will discuss that it is possible to implement a gradient descent in infinite dimensions by using **gradient flows** on a metric space of probability measures (Ambrosio et al., 2005). More specifically, one can solve the optimisation problem (1) by following the curve of steepest descent in the 2-Wasserstein space. As it turns out, this approach is not only theoretically sound, but also conceptually elegant: it unifies existing algorithms for uncertainty quantification in deep learning, and even allows us to derive new ones. We refer to algorithms based on some form of infinite-dimensional gradient descent as **infinite-dimensional GVI (ID-GVI)**. Infinite-dimensional gradient descent methods have recently gained attention in the machine learning community. For existing methods of this kind, the goal is to generate samples from a target $\widehat{Q}$ that has a *known* form (such as the Bayes posterior) by applying a gradient flow to $Q \in \mathcal{P}(\mathbb{R}^J) \mapsto E(Q, \widehat{Q})$ where $E(\cdot, \cdot)$ is a discrepancy measure. Some methods apply the Wasserstein gradient flow (WGF) for different choices of $E$ (Arbel et al., 2019; Korba et al., 2021; Glaser et al., 2021), whilst other methods like Stein variational gradient descent (SVGD) (Liu and Wang, 2016) stay within the Bayesian paradigm ($E = \mathrm{KL}$, $\widehat{Q}$ = Bayes posterior) but use a gradient flow other than the WGF (Liu, 2017). D'Angelo and Fortuin (2021) exploit the WGF in the standard Bayesian context and combine it with different gradient estimators (Li and Turner, 2017; Shi et al., 2018) to obtain repulsive deep ensembling schemes. Note that this is different from the repulsive approach we introduce later in this paper: Our repulsion term is the consequence of a regulariser, not a gradient estimator. Since our focus is on tackling the problems associated with non-convex optimisation in Euclidean space, the approach we propose is inherently different from all of these existing methods: our target $Q^*$ is only implicitly defined via (1), and not known explicitly.

## 3 Gradient flows in finite and infinite dimensions

Before we can realise our ambition to solve (1) with an ID-GVI scheme, we need to cover the relevant bases. To this end, we will discuss gradient flows in finite and infinite dimensions, and explain how they can be used to construct infinite-dimensional gradient descent schemes. In essence, a gradient flow is the limit of a gradient descent whose step size goes to zero. While the current section introduces this idea for the finite-dimensional case for ease of exposition, its use in constructing algorithms within the current paper will be for the infinite-dimensional case.

Gradient descent finds local minima of losses $\ell : \mathbb{R}^J \to \mathbb{R}$ by iteratively improving an initial guess $\theta_0 \in \mathbb{R}^J$ through the update $\theta_{k+1} := \theta_k - \eta \nabla \ell(\theta_k)$, $k \in \mathbb{N}$, where $\eta > 0$ is a step-size and $\nabla \ell$ denotes the gradient of $\ell$. For sufficiently small $\eta > 0$, this update can equivalently be written as

$$\theta_{k+1} = \underset{\theta \in \mathbb{R}^J}{\arg\min} \left\{ \ell(\theta) + \frac{1}{2\eta} \|\theta - \theta_k\|_2^2 \right\}. \tag{2}$$

Gradient flows formalise the following logic: for any fixed $\eta$, we can continuously interpolate the corresponding gradient descent iterates $\{\theta_k\}_{k\in\mathbb{N}_0}$. To do this, we simply define a function $\theta^\eta : [0,\infty) \to \mathbb{R}$ as $\theta^\eta(t) := \theta_{t/\eta}$ for $t \in \eta\mathbb{N}_0 := \{0,\eta,2\eta,...\}$. For $t \notin \eta\mathbb{N}_0$ we linearly interpolate[3]. As $\eta \to 0$, the function $\theta^\eta$ converges to a differentiable function $\theta_* : [0,\infty) \to \mathbb{R}$ called the gradient flow of $\ell$, because it is characterised as solution to the ordinary differential equation (ODE) $\theta'_*(t) = -\nabla\ell(\theta_*(t))$ with initial condition $\theta_*(0) = \theta_0$. Intuitively, $\theta_*(t)$ is a continuous-time version of discrete-time gradient descent; and navigates through the loss landscape so that at time $t$, an infinitesimally small step in the direction of steepest descent is taken. Put differently: gradient descent is nothing but an Euler discretisation of the gradient flow ODE (see also Santambrogio, 2017). The result is that for mathematical convenience, one often analyses discrete-time gradient descent as though it were a continuous gradient flow—with the hope that for sufficiently small $\eta$, the behaviour of both will essentially be the same.

Our results in the infinite-dimensional case follow this principle: we propose an algorithm based on discretisation, but use continuous gradient flows to guide the analysis. To this end, the next section generalises gradient flows to the nonlinear infinite-dimensional setting.

### 3.1 Gradient flows in Wasserstein spaces

Let $\mathcal{P}_2(\mathbb{R}^J)$ be the space of probability measures with finite second moment equipped with the 2-Wasserstein metric given as

$$W_2(P,Q)^2 = \inf\left\{\int ||\theta - \theta'||_2^2\, d\pi(\theta,\theta') : \pi \in \mathcal{C}(P,Q)\right\}$$

where $\mathcal{C}(P,Q) \subset \mathcal{P}(\mathbb{R}^J \times \mathbb{R}^J)$ denotes the set of all probability measure on $\mathbb{R}^J \times \mathbb{R}^J$ such that $\pi(A \times \mathbb{R}^J) = P(A)$ and $\pi(\mathbb{R}^J \times B) = Q(B)$ for all $A,B \subset \mathbb{R}^J$ (see also Chapter 6 of Villani et al., 2009). Further, let $L : \mathcal{P}_2(\mathbb{R}^J) \to (-\infty,\infty]$ be some functional—for example $L$ in (1). In direct analogy to (2), we can improve upon an initial guess $Q_0 \in \mathcal{P}_2(\mathbb{R}^J)$ by iteratively solving

$$Q_{k+1} := \underset{Q\in\mathcal{P}_2(\mathbb{R}^J)}{\arg\min}\left\{L(Q) + \frac{1}{2\eta}W_2(Q,Q_k)^2\right\}$$

for $k \in \mathbb{N}_0$ and small $\eta > 0$ (see Chapter 2 of Ambrosio et al., 2005, for details). Again, for $\eta \to 0$, an appropriate limit yields a continuously indexed family of measures $\{Q(t)\}_{t\geq 0}$. If $L$ is sufficiently smooth and $Q_0 = Q(0)$ has Lebesgue density $q_0$, the time evolution for the corresponding pdfs $\{q(t)\}_{t\geq 0}$ is given by the partial differential equation (PDE)

$$\partial_t q(t,\theta) = \nabla \cdot \Big(q(t,\theta)\,\nabla_W L\big[Q(t)\big](\theta)\Big), \tag{3}$$

with $q(0,\cdot) = q_0$ (Villani, 2003, Section 9.1). Here $\nabla \cdot f := \sum_{j=1}^J \partial_j f_j$ denotes the divergence operator and $\nabla_W L[Q] : \mathbb{R}^J \to \mathbb{R}^J$ the Wasserstein gradient (WG) of $L$ at $Q$. For the purpose of this paper, it is sufficient to think of the WG as a gradient of the first variation; i.e. $\nabla_W L[Q] = \nabla L'[Q]$ where $L'[Q] : \mathbb{R}^J \to \mathbb{R}^J$ is the first variation of $L$ at $Q$ (Villani et al., 2009, Exercise 15.10). The Wasserstein gradient flow (WGF) for $L$ is then the solution $q^\dagger$ to the PDE (3). If $L$ is chosen as in (1), our hope is that the logic of finite-dimensional gradient descent carries over; and that $\lim_{t\to\infty} q^\dagger(t,\cdot)$ is in fact the density $q^*$ corresponding to $Q^*$.

Following this reasoning, this paper applies the WGF for (1) to obtain an ID-GVI algorithm. In Section 4 and Appendices C–F, we formally show that the WGF indeed yields $Q^*$ (in the limit as $t \to \infty$) for a number of regularisers of practical interest.

### 3.2 Realising the Wasserstein gradient flow

In theory, the PDE in (3) could be solved numerically in order to implement the infinite-dimensional gradient descent for (1). In practice however, this is impossible: numerical solutions to PDEs become computationally infeasible for the high-dimensional parameter spaces which are common in deep learning applications. Rather than trying to first approximate the $q$ solving (3) and then sampling from its limit in a second step, we will instead formulate equations which replicate how the

---

[3]This means $\theta^\eta(t) := \frac{\theta_{s+1}-\theta_s}{\eta}\big(t - s\eta\big) + \theta_s$, for $t \in [s\eta, (s+1)\eta)$ and $s \in \mathbb{N}_0$.

samples from the solution to (3) evolve in time. This leads to tractable inference algorithms that can be implemented in high dimensions.

Given the goal of producing samples directly, we focus on a particular form of loss that is well-studied in the context of thermodynamics (Santambrogio, 2015, Chapter 7), and which recovers various forms of the GVI problem in (1) (see Section 4). In thermodynamics, $Q \in \mathcal{P}_2(\mathbb{R}^J)$ describes the distribution of particles located at specific points in $\mathbb{R}^J$. The overall energy of a collection of particles sampled from $Q$ is decomposed into three parts: (i) the external potential $V(\theta)$ which acts on each particle individually, (ii) the interaction energy $\kappa(\theta, \theta')$ describing pairwise interactions between particles, and (iii) an overall entropy of the system measuring how concentrated the distribution $Q$ is. Taking these components together, we obtain the so called **free energy**

$$L^{\mathrm{fe}}(Q) := \int V(\theta)\, dQ(\theta) + \frac{\lambda_1}{2} \iint \kappa(\theta, \theta')\, dQ(\theta) dQ(\theta') + \lambda_2 \int \log q(\theta) q(\theta)\, d\theta, \quad (4)$$

for $Q \in \mathcal{P}_2(\mathbb{R}^J)$ with Lebesgue density $q$, $\lambda_1 \geq 0$, $\lambda_2 \geq 0$. Note that for $\lambda_2 > 0$ we implicitly assume that $Q$ has a density. Following Section 9.1 in Villani et al. (2009), its WG is

$$\nabla_W L^{\mathrm{fe}}[Q](\theta) = \nabla V(\theta) + \lambda_1 \int (\nabla_1 \kappa)(\theta, \theta')\, dQ(\theta') + \lambda_2 \nabla \log q(\theta),$$

where $\theta \in \mathbb{R}^J$, and $\nabla_1 \kappa$ denotes the gradient of $\kappa$ with respect to the first variable. We plug this into (3) to obtain the desired density evolution. Importantly, this time evolution has the exact form of a nonlinear Fokker-Planck equation associated with a stochastic process of McKean-Vlasov type (see Appendix B for details). Fortunately for us, it is well-known that such processes can be approximated through interacting particles (Veretennikov, 2006) generated by the following procedure:

**Step 1:** Sample $N_E \in \mathbb{N}$ particles $\theta_1(0), \ldots, \theta_{N_E}(0)$ independently from $Q_0 \in \mathcal{P}_2(\mathbb{R}^J)$.

**Step 2:** Evolve the particle $\theta_n$ by following the stochastic differential equation (SDE)

$$d\theta_n(t) = -\Big( \nabla V\big(\theta_n(t)\big) + \frac{\lambda_1}{N_E} \sum_{j=1}^{N_E} (\nabla_1 \kappa)\big(\theta_n(t), \theta_j(t)\big) \Big) dt + \sqrt{2\lambda_2} dB_n(t), \quad (5)$$

for $n = 1, \ldots, N_E$, and $\{B_n(t)\}_{t>0}$ stochastically independent Brownian motions.

As $N_E \to \infty$, the distribution of $\theta_1(t), \ldots, \theta_{N_E}(t)$ evolves in $t$ in the same way as the sequence of densities $q(t, \cdot)$ solving (3). This means that we can implement infinite-dimensional gradient descent by following the WGF and simulating trajectories for infinitely many interacting particles according to the above procedure. In practice, we can only simulate finitely many trajectories over a finite time horizon. This produces samples $\theta_1(T), \ldots, \theta_{N_E}(T)$ for $N_E \in \mathbb{N}$ and $T > 0$. Our intuition and Section 4 tell us that, as desired, the distribution of $\theta_1(T), \ldots, \theta_{N_E}(T)$ will be close to the global minimiser of $L^{\mathrm{fe}}$.

In the next section, we will use the above algorithm to construct an ID-GVI method producing samples approximately distributed according to $Q^*$ defined in (1). Since $N_E$ and $T$ are finite, and since we need to discretise (131), there will be an approximation error. Given this, how good are the samples produced by such methods? As we shall demonstrate in Section 5, the approximation errors are small, and certainly should be expected to be much smaller than those of standard VI and other FD-GVI methods.

## 4 Optimisation in the space of probability measures

With the WGF on thermodynamic objectives in place, we can now finally show how it yields ID-GVI algorithms to solve (1). We put particular focus on the analysis of the regulariser $D$; providing new perspectives on heuristics for uncertainty quantification in deep learning in the process. Specifically, we establish formal links explaining how they may (not) be understood as a Bayesian procedure. Beyond that, we derive the WGF associated with regularisation using the maximum mean discrepancy, and provide a theoretical analysis of its convergence properties.

## 4.1 Unregularised probabilistic lifting: Deep ensembles

We start the analysis with the base case of an unregularised functional $L(Q) = \int \ell(\theta)dQ(\theta)$, corresponding to $\lambda = 0$ in (1). This is also a special case of (4) with $\lambda_1 = \lambda_2 = 0$. As $\lambda_1 = 0$, there is no interaction term, and all particles can be simulated independently from one another as

$$\theta_1(0), \ldots, \theta_{N_E}(0) \sim Q_0, \quad \theta'_n(t) = -\nabla\ell(\theta_n(t)), \quad n = 1, \ldots, N_E.$$

This simple algorithm happens to coincide exactly with how deep ensembles (DEs) are constructed (see e.g. Lakshminarayanan et al., 2017). In other words: the simple heuristic of running gradient descent algorithm several times with random initialisations sampled from $Q_0$ is an approximation of the WGF for the *unregularised* probabilistic lifting of the loss function $\ell$.

Following the WGF in this case does not generally produce samples from a global minimiser of $L$. Indeed, the fact that $L$ generally does not even have a unique global minimiser was the motivation for regularisation in (1). Even if $L$ had a unique minimiser however, a DE would not find it. The result below proves this formally: unsurprisingly, deep ensembles simply sample the local minima of $\ell$ with a probability that depends on the domain of attraction and the initialisation distribution $Q_0$.

**Theorem 1.** *If $\ell$ has countably many local minima $\{m_i : i \in \mathbb{N}\}$, then it holds independently for each $n = 1, \ldots, N_E$ that*

$$\theta_n(t) \xrightarrow{\mathcal{D}} \sum_{i=1}^{\infty} Q_0(\Theta_i)\,\delta_{m_i} =: Q_\infty$$

*for $t \to \infty$. Here $\xrightarrow{\mathcal{D}}$ denotes convergence in distribution and $\Theta_i = \{\theta \in \mathbb{R}^J : \lim_{t\to\infty}\theta_*(t) = m_i \text{ and } \theta_*(0) = \theta\}$ denotes the domain of attraction for $m_i$ with respect to the gradient flow $\theta_*$.*

A proof with technical assumptions—most importantly a version of the famous Lojasiewicz inequality—is in Appendix C. Theorem 1 derives the limiting distribution for $\theta_1(T), \ldots, \theta_{N_E}(T)$, which shows that—unless all local minima are global minima—the WGF does not generate samples from a global minimum of $Q \mapsto L(Q)$ for the unregularised case $\lambda = 0$. Note that the conditions of this result simplify the situation encountered in deep learning, where the set of minimisers would typically be uncountable (Liu et al., 2022). While one could derive a very similar result for the case of uncountable minimisers, this becomes notationally cumbersome and would obscure the main point of the Theorem—that $Q_\infty$ strongly depends on the initialisation $Q_0$. Importantly, the dependence of $Q_\infty$ on $Q_0$ remains true for all losses constructed via deep learning architectures. However, despite these theoretical shortcomings, DEs remain highly competitive in practice and typically beat FD-GVI methods like standard VI (Ovadia et al., 2019; Fort et al., 2019). This is perhaps not surprising: DEs implement an infinite-dimensional gradient descent, while FD-GVI methods are parametrically constrained. Perhaps more surprisingly, we observe in Section 5 that DEs can even easily compete with the more theoretically sound and regularised ID-GVI methods that will be discussed in Section 4.2 and 4.3. We study this phenomenon in Section 5, and find that it is a consequence of the fact that in deep learning, $N_E$ is small compared to the number of local minima (cf. Figure 4).

## 4.2 Regularisation with the Kullback-Leibler divergence: Deep Langevin ensembles

In Section 2, we argued for regularisation by $D$ to ensure a unique minimiser $Q^*$. The Kullback-Leibler divergence ($D = \text{KL}$) is the canonical choice for (generalised) Bayesian and PAC-Bayesian methods (Bissiri et al., 2016; Knoblauch et al., 2022; Guedj and Shawe-Taylor, 2019; Alquier, 2021b). Now, $Q^*$ has a known form: if $P$ has a pdf $p$, it has an associated density given by $q^*(\theta) \propto \exp(-\frac{1}{\lambda}\ell(\theta))p(\theta)$ (Knoblauch et al., 2022, Theorem 1).

Notice that the KL-regularised version of $L$ in (1) can be rewritten in terms of the objective $L^{\text{fe}}$ in (4) by setting $V(\theta) = \ell(\theta) - \lambda \log p(\theta)$, $\lambda_1 = 0$ and $\lambda_2 = \lambda$. Compared to the unregularised objective of the previous section (where $\lambda = 0$), the external potential is now adjusted by $-\lambda \log p(\theta)$, forcing $Q^*$ to allocate more mass in regions where $p$ has high density. Beyond that, the presence of the negative entropy term has three effects: it ensures that the objective is strictly convex, that $Q^*$ is more spread out, and that it has a density $q^*$. Since $\lambda_1 = 0$, the corresponding particle method still does not have an interaction and is given as

$$\theta_1(0), \ldots, \theta_{N_E}(0) \sim Q_0 \quad d\theta_n(t) = -\nabla V(\theta_n(t))dt + \sqrt{2\lambda}dB_n(t), \quad n = 1, \ldots, N_E. \quad (6)$$

Clearly, this is just the Langevin SDE and we call this approach the **deep Langevin ensemble (DLE)**. While the name may suggest that DLE is equivalent to the unadjusted Langevin algorithm (ULA) (Roberts and Tweedie, 1996), this is not so: for $T$ discretisation steps $t_1, t_2, \ldots t_T$, DLE approximates measures using the *end-points* of $N_E$ trajectories given by $\{\theta_n(t_T)\}_{n=1}^{N_E}$. In contrast, ULA would use a (sub)set of the samples $\{\theta_1(t_i)\}_{i=1}^{T}$ generated from one single particle's *trajectory*. To analyse DLEs, we build on the Langevin dynamics literature: in Appendix D, we show that $\theta_n(t) \xrightarrow{\mathcal{D}} Q^*$ as $t \to \infty$, independently for each $n = 1, \ldots, N_E$. Hence $\theta_1(T), \ldots, \theta_{N_E}(T)$ will for large $T > 0$ be approximately distributed according to $Q^*$. Comparing DE and DLE in this light, we note several important key differences: $Q^*$ as defined per (1) is unique, has the form of a Gibbs measure, and can be sampled from using (6). In contrast, unregularised DE produces samples from $Q_\infty$ in Theorem 1 which is not the global minimiser. Specifically neither $Q_\infty$ nor $Q^*$ for DEs correspond to the Bayes posterior. It is therefore not a Bayesian procedure in any commonly accepted sense of the word.

### 4.3 Regularisation with maximum mean discrepancy: Deep repulsive Langevin ensembles

Regularising with KL is attractive because $Q^*$ has a known form. However, in our theory, there is no reason to restrict attention to a single type of regulariser: we introduced $D$ to convexify our objective. It is therefore of theoretical and practical interest to see which algorithmic effects are induced by other regularisers. We illustrate this by first considering regularisation using the squared maximum-mean discrepancy (MMD) (see e.g. Gretton et al., 2012) only, and then a combination of MMD and KL.

For a kernel $\kappa : \mathbb{R}^J \times \mathbb{R}^J \to \mathbb{R}$, the squared MMD between measures $Q$ and $P$ is

$$\text{MMD}(Q, P)^2 = \iint \kappa(\theta, \theta') dQ(\theta) dQ(\theta') - 2 \iint \kappa(\theta, \theta') dQ(\theta) dP(\theta')$$
$$+ \iint \kappa(\theta, \theta') dP(\theta) dP(\theta').$$

MMD measures the difference between within-sample similarity and across-sample similarity, so it is smaller when samples from $P$ are similar to samples from $Q$, but also larger when samples within $Q$ are similar to each other. This means that regularising (1) with $D = \text{MMD}^2$ introduces interactions characterised precisely by the kernel $\kappa$, and we can show this explicitly by rewriting $L$ of (1) into the form of $L^{\text{fe}}$ in (4). In other words, inclusion of $\text{MMD}^2$ makes particles repel each other, making it more likely that they fall into different (rather than the same) local minima. Writing the kernel mean embedding as $\mu_P(\theta) := \int \kappa(\theta, \theta') dP(\theta')$, we see that up to a constant not depending on $Q$, $L(Q) = L^{\text{fe}}(Q)$ for $V(\theta) = \ell(\theta) - \lambda_1 \mu_P(\theta)$, $\lambda = \frac{\lambda_1}{2}$, and $\lambda_2 = 0$. While we can show that a global minimiser $Q^*$ exists, and while we could produce particles using the algorithm of Section 3.2, we cannot guarantee that they are distributed according to $Q^*$ (see Appendix F). Essentially, this is because in certain situations, we cannot guarantee that $Q^*$ has a density for $D = \text{MMD}^2$.

To remedy this problem, we additionally regularise with the KL: since $\text{KL}(Q, P) = \infty$ if $P$ has a Lebesgue density but $Q$ has not, this now guarantees that $Q^*$ has a density $q^*$. In terms of (1), this means that $D = \lambda \text{MMD}^2 + \lambda' \text{KL}$. Adding regularisers like this has a long tradition, and is usually done to combine the different strengths of various regularisers (see e.g. Zou and Hastie, 2005). Here, we follow this logic: the KL ensures that $Q^*$ has a density, and the MMD makes particles repel each other. With this, we can rewrite $L(Q)$ in terms of $L^{\text{fe}}(Q)$ up to a constant not depending on $Q$ by taking $\lambda = \frac{\lambda_1}{2}$, $\lambda' = \lambda_2$, and $V(\theta) = \ell(\theta) - \lambda_1 \mu_P(\theta) - \lambda_2 \log p(\theta)$. Using the same algorithmic blueprint as before, we evolve particles according to (5). As these particles follow an augmented Langevin SDE that incorporates repulsive particle interactions via $\kappa$, we call this method the **deep repulsive Langevin ensemble (DRLE)**. We show in Theorem (2) (cf. Appendix E for details and assumptions) that DRLEs generate samples from the global minimiser $Q^*$ in the infinite particle and infinite time horizon limit.

**Theorem 2.** *Let $Q^{n,N_E}(T)$ be the distribution of $\theta_n(T)$, $n = 1, \ldots, N_E$, generated via (5). Then $Q^{n,N_E}(T) \xrightarrow{\mathcal{D}} Q^*$ for each $n = 1, \ldots, N_E$ and as $N_E \to \infty$, $T \to \infty$.*

This is remarkable: we have constructed an algorithm that generates samples from the global minimiser $Q^*$—even though a formal expression for what exactly $Q^*$ looks like is unknown! This

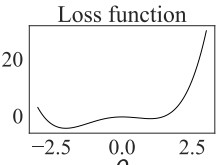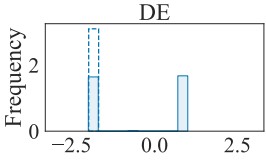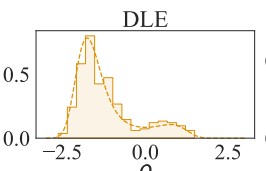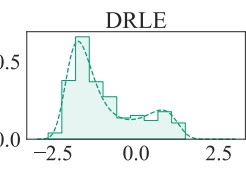

Figure 2: We generate $N_E = 300$ particles from DE, DLE and DRLE. The theoretically optimal global minimisers $Q^*$ are depicted with dashed line strokes, and the generated samples are displayed via histograms. We use $P = Q_0 = \mathcal{N}(0, 1)$ for DLE and DRLE. Notice that the optimal $Q^*$ differs slightly between DLE and DRLE.

demonstrates how impressively powerful the WGF is as tool to derive inference algorithms. Note that this is completely different from sampling methods employed for Bayesian methods, for which the form of $Q^*$ is typically known explicitly up to a proportionality constant.

A notable shortcoming of Theorem 2 is its asymptotic nature. A more refined analysis could quantify how fast the convergence happens in terms of $N_E, T$, the SDE's discretisation error, and maybe even the estimation errors due to sub-sampling of losses for constructing gradients. While the existing literature could be adapted to derive the speed of convergence for DRLE in $T$ (Ambrosio et al., 2005, Section 11.2), this would require a strong convexity assumption on the potential $V$, which will not be satisfied for any applications in deep learning. This is perhaps unsurprising: even for the Langevin algorithm—probably the most thoroughly analysed algorithm in this literature—no convergence rates have been derived that are applicable to the highly multi-modal target measures encountered in Bayesian deep learning (Wibisono, 2019; Chewi et al., 2022). That being said, for the case of deep learning, FD-GVI approaches fail to provide even the most basic asymptotic convergence guarantees. Thus, the fact that it is even possible for us to provide *any* asymptotic guarantees derived from realistic assumptions marks a significant improvement over the available theory for FD-GVI methods, and—by virtue of Theorem 1—over DEs as well.

## 5   Experiments

Since the paper's primary focus is on theory, we use two experiments to reinforce some of the predictions it makes in previous sections, and a third experiment that shows why–in direct contradiction to a naive interpretation of the presented theory–it is typically difficult to beat simple DEs. More details about the conducted experiments can be found in Appendix G. The code is available on `https://github.com/sghalebikesabi/GVI-WGF`.

**Global minimisers:** Figure 2 illustrates the theory of Sections 4 and Appendices C–F: DLE and DRLE produce samples from their respective global minimisers, while DE produces a distribution which—-in accordance with Theorem 1—does not correspond to the global minimiser of $Q \mapsto \int \ell(\theta) \, dQ(\theta)$ over $\mathcal{P}(\mathbb{R}^J)$ (which is given as Dirac measure located at $\theta = -2$).

**FD-GVI vs ID-GVI:** Figure 3 illustrates two aspects. First, the effect of regularisation for DLE and DRLE is that particles spread out around the local minima. In comparison, DE particles fall directly into the local minima. Second, FD-GVI (with Gaussian parametric family) leads to qualitatively poorer approximations of $Q^*$. This is because the ID-GVI methods explore the whole space $\mathcal{P}_2(\mathbb{R}^J)$, whilst FD-GVI is limited to learning a unimodal Gaussian.

**DEs vs D(R)LEs, and why finite $N_E$ matters:** Table 1 compares DE, DLE and DRLE on a number of real world data sets, and finds a rather random distribution of which method performs best. This seems to contradict our theory, and suggests there is essentially no difference between regularised and unregularised ID-GVI. What explains the discrepancy? Essentially, it is the fact that $N_E$ is not only finite, but much smaller than the number of minima found in the loss landscape of deep learning. In this setting, each particle moves into the neighbourhood of a well-separated single local minimum and typically never escapes, even for very large $T$. We illustrate this in Figure 4 with a toy example. We choose a uniform prior $P$ and initialisation $Q_0$ and the loss $\ell(\theta) := -|\sin(\theta)|$, $\theta \in [-1000\pi, 1000\pi]$, which has 2000 local minima. Correspondingly $Q^*$ will have many local modes for all methods. Note that $\nabla V$ is the same for all approaches since $\log p$ and $\mu_P$ are constant. The difference between the methods boils down to repulsive and noise effects. However, these noise effects are not significant if each particle is stuck in a single mode: the particles will bounce around

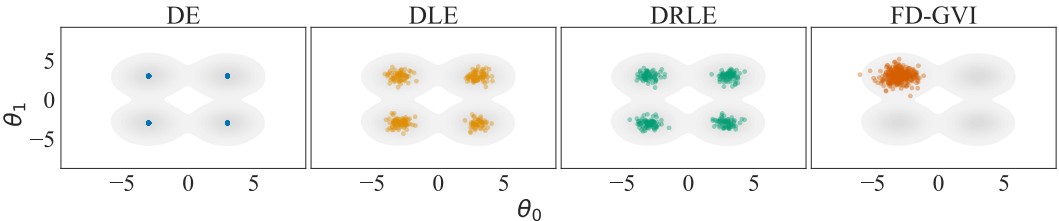

Figure 3: We generate $N_E = 300$ particles from DE, DLE, DRLE and FD-GVI with Gaussian parametrisation. The multimodal loss $\ell$ is plotted in grey and the particles of the different methods are layered on top. The prior in this example is flat, i.e. $\log p$ and $\mu_P$ are constant. The initialisation $Q_0$ is standard Gaussian.

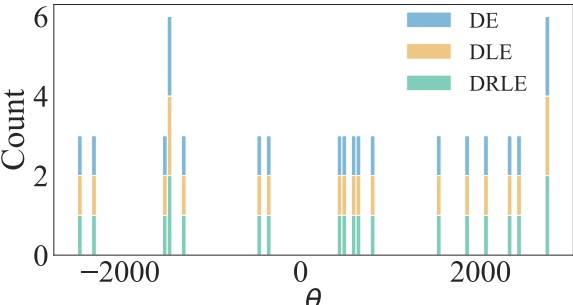

Figure 4: We generate $N_E = 20$ samples from the three infinite-dimensional gradient descent procedures discussed in Section 4. The $x$-axis shows the location of the particles after training. Since the same initialisation $\theta_n(0)$ is chosen for all methods, we observe that particles fall into the same local modes. Further, 16/20 particles are alone in their respective local modes and the location of the particles varies only very little between the different methods (which is why they are in the same bucket in the above histogram). See also Figure 5 in the Appendix for an alteration of Figure 3 with only 4 particles which emphasizes the same point.

their local modes, but not explore other parts of the space. This implies that they will not improve the approximation quality of $Q^*$. Note that this problem is a direct parallel to multi-modality—a well-known problem for Markov Chain Monte Carlo methods (see e.g. Syed et al., 2022).

## 6    Conclusion

In this paper, we used infinite-dimensional gradient descent via Wasserstein gradient flows (WGFs) (see e.g. Ambrosio et al., 2005) and the lens of generalised variational inference (GVI) (Knoblauch et al., 2022) to unify a collection of existing algorithms under a common conceptual roof. Arguably, this reveals the WGF to be a powerful tool to analyse ensemble methods in deep learning and beyond. Our exposition offers a fresh perspective on these methodologies, and plants the seeds for new ensemble algorithms inspired by our theory. We illustrated this by deriving a new algorithm that includes a repulsion term, and use our theory to prove that ensembles produced by the algorithm converge to a global minimum. A number of experiments showed that the theory developed in the current paper is useful, and showed why the performance difference between simple deep ensembles and more intricate schemes may not be numerically discernible for loss landscapes with many local minima.

| | KIN8NM | CONCRETE | ENERGY | NAVAL | POWER | PROTEIN | WINE | YACHT |
|---|---|---|---|---|---|---|---|---|
| DE | $\mathbf{0.33_{\pm0.1}}$ | $6.10_{\pm0.3}$ | $2.83_{\pm0.2}$ | $-0.40_{\pm0.3}$ | $\mathbf{13.70_{\pm2.6}}$ | $\mathbf{11.22_{\pm2.2}}$ | $14.65_{\pm1.9}$ | $2.20_{\pm0.4}$ |
| DLE | $13.25_{\pm4.3}$ | $\mathbf{5.11_{\pm0.2}}$ | $\mathbf{2.43_{\pm0.1}}$ | $3.46_{\pm2.4}$ | $13.87_{\pm2.3}$ | $43.20_{\pm12.5}$ | $13.73_{\pm1.4}$ | $\mathbf{1.64_{\pm0.1}}$ |
| DRLE | $0.46_{\pm0.1}$ | $8.30_{\pm0.6}$ | $4.01_{\pm0.3}$ | $\mathbf{-3.04_{\pm0.2}}$ | $23.21_{\pm2.0}$ | $48.80_{\pm2.1}$ | $\mathbf{7.13_{\pm0.6}}$ | $7.80_{\pm2.7}$ |

Table 1: Table compares the average (Gaussian) negative log likelihood in the test set for the three ID-GVI methods on some UCI-regression data sets (Lichman, 2013). We observe that no method consistently outperforms any of the others.

## Acknowledgements

VW was supported by the scatchered scholarship and the EPSRC grant EP/W005859/1. SG was a PhD student of the EPSRC CDT in Modern Statistics and Statistical Machine Learning (EP/S023151/1), and also received funding from the Oxford Radcliffe Scholarship and Novartis. JK was funded by EPSRC grant EP/W005859/1.

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

# A Existence and uniqueness of global minimiser

In this section, we discuss assumptions under which the global minimiser of the optimisation problem

$$L(Q) = \int \ell(\theta) \, dQ(\theta) + \lambda D(Q, P) \tag{7}$$

over $\mathcal{P}(\mathbb{R}^J)$ exists and is unique. We assume throughout that the optimisation problem is not pathological, in the sense that there exists a measure $\widehat{Q} \in \mathcal{P}(\mathbb{R}^J)$ such that $L(\widehat{Q}) < \infty$. This is in applications often trivial to verify. A good candidate for $\widehat{Q}$ is typically the reference measure $P$.

**Loss assumptions** Let $\ell : \mathbb{R}^J \to \mathbb{R}$ be a loss satisfying the following assumptions:

(L1) The loss $\ell$ is bounded from below which means that

$$c := \inf \left\{ \ell(\theta) : \theta \in \mathbb{R}^J \right\} > -\infty. \tag{8}$$

(L2) The loss is norm-coercive which means that

$$\ell(\theta) \to \infty \tag{9}$$

if $\|\theta\| \to \infty$.

(L3) The loss $\ell$ is lower semi-continuous which means that

$$\liminf_{\theta \to \theta_0} \ell(\theta) \geq \ell(\theta_0) \tag{10}$$

for all $\theta_0 \in \mathbb{R}^J$.

**Regulariser assumptions** Let $D : \mathcal{P}(\mathbb{R}^J) \times \mathcal{P}(\mathbb{R}^J) \to [0, \infty]$ be a regulariser and $P \in \mathcal{P}(\mathbb{R}^J)$ a reference measure. We define $D_P(\cdot) := D(\cdot, P)$ for notational convenience. We assume the following for $D_P$:

(D1) The function $D_P$ is lower semi-continuous w.r.t. to the topology of weak-convergence, i.e. for all sequences $(Q_n)_{n \in \mathbb{N}} \subset \mathcal{P}(\mathbb{R}^J)$ and all $Q$ with $D_P(Q) < \infty$, it holds that $Q_n \xrightarrow{\mathcal{D}} Q$ implies

$$\liminf_{n \to \infty} D_P(Q_n) \geq D_P(Q). \tag{11}$$

Here, $\xrightarrow{\mathcal{D}}$ denotes convergence in distribution.

(D2) $D_P$ is strictly convex, i.e. for all $Q_1 \neq Q_2 \in \mathcal{P}(\mathbb{R}^J)$ with $D_P(Q_1) < \infty$ and $D_P(Q_2) < \infty$, it holds that

$$D_P\big(\alpha Q_1 + (1 - \alpha)Q_2\big) < \alpha D_P(Q_1) + (1 - \alpha)D_P(Q_2) \tag{12}$$

with $\alpha \in (0, 1)$.

The next theorem provides an existence result for the optimisation problem (7). The result is similar in spirit to Lemma 2.1 in Knoblauch (2021) with the important difference that our assumptions are easier to verify, since they are formulated in terms of $\ell$ and $D_P$.

**Theorem 3** (Existence of global minimiser). *Under the assumptions (L1)-(L3) and (D1) there exists a probability measure $Q^* \in \mathcal{P}(\mathbb{R}^J)$ with*

$$L(Q^*) = \inf \left\{ L(Q) : Q \in \mathcal{P}(\mathbb{R}^J) \right\}. \tag{13}$$

*Proof.* Let $c > -\infty$ be the lower bound for $\ell$. It follows immediately that $L(Q) \geq c$ for all $Q \in \mathcal{P}(\mathbb{R}^J)$ since $D(P, Q) \geq 0$. As a consequence we know that

$$\infty > L^* := \inf \left\{ L(Q) : Q \in \mathcal{P}(\mathbb{R}^J) \right\} \geq c > -\infty. \tag{14}$$

By definition of the infimum we can construct a sequence $l_n = L(Q_n) \in \mathbb{R}$ in the image of $L$ such

$$l_n \to L^* \tag{15}$$

for $n \to \infty$. We now show by contradiction that the corresponding sequence $(Q_n) \subset \mathcal{P}(\mathbb{R}^J)$ is *tight*[4]. Assume that $(Q_n)$ is not tight. By definition we can then find an $\epsilon > 0$ such that for each $k \in \mathbb{N}$ there exists $n = n_k \in \mathbb{N}$ with $Q_{n_k}([-k,k]^J) \leq 1 - \epsilon$. We set $A_k := [-k,k]^J \subset \mathbb{R}^J$ and obtain

$$l_{n_k} = L(Q_{n_k}) \tag{16}$$

$$= \int_{A_k} \ell(\theta)\, dQ_{n_k}(\theta) + \int_{\mathbb{R}^J \setminus A_k} \ell(\theta)\, dQ_{n_k}(\theta) + \lambda D(Q, P) \tag{17}$$

$$\geq \int_{A_k} \ell(\theta)\, dQ_{n_k}(\theta) + \int_{\mathbb{R}^J \setminus A_k} \ell(\theta)\, dQ_{n_k}(\theta) \tag{18}$$

$$\geq c Q_{n_k}(A_k) + \inf\left\{\ell(\theta) : \theta \in \mathbb{R}^J \setminus A_k\right\} Q_{n_k}(\mathbb{R}^J \setminus A_k) \tag{19}$$

$$\geq c Q_{n_k}(A_k) + \epsilon \inf\left\{\ell(\theta) : \theta \in \mathbb{R}^J \setminus A_k\right\}. \tag{20}$$

Due to the coerciveness of $\ell$, we know that $\inf\left\{\ell(\theta) : \theta \in \mathbb{R}^J \setminus A_k\right\} \to \infty$ for $k \to \infty$ and therefore $l_{n_k} \to \infty$ for $k \to \infty$. However, this is a contradiction: The sequence $(l_n)$ is convergent and therefore in particular bounded. As a consequence, it cannot contain the unbounded sub-sequence $(l_{n_k})$. It follows that the sequence $(Q_n)$ is tight. By Prokhorov's theorem we can now extract a sub sequence $(Q_{n_k})$ of $(Q_n)$ and a measure $Q^* \in \mathcal{P}(\mathbb{R}^J)$ such that

$$Q_{n_k} \xrightarrow{\mathcal{D}} Q^* \tag{21}$$

for $k \to \infty$. Due to Lemma 5.1.7 in Ambrosio et al. (2005) the lower semi-continuity of $\ell$ implies that $Q \mapsto \int \ell(\theta)\, dQ(\theta)$ is lower semi-continuous. This combined with the lower semi-continuity of $D_P$ gives

$$\liminf_{k \to \infty} L(Q_{n_k}) \geq L(Q^*). \tag{22}$$

From this it immediately follows that

$$L(Q^*) \leq \liminf_{k \to \infty} L(Q_{n_k}) = L^*, \tag{23}$$

but by definition $L^*$ is the global minimum of $L$ which implies $L^* \leq L(Q^*)$. We therefore conclude that $L(Q^*) = L^*$. $\qquad\square$

Theorem 3 only shows the existence of a global minimiser. In order to show uniqueness we use the convexity assumption (D2). The proof is the same as in finite dimensions and only included for completeness.

**Theorem 4** (Uniqueness of global minimiser). *Assume that (D2) holds. Then, the global minimiser of $L$ is unique (whenever it exists).*

*Proof.* Assume there exits two probability measures $Q_1, Q_2 \in \mathcal{P}(\mathbb{R}^J)$ such that

$$L(Q_1) = L^* = L(Q_2). \tag{24}$$

where $\infty > L^* := \inf\left\{L(Q) : Q \in \mathcal{P}(\mathbb{R}^J)\right\} > -\infty$. We define the probability measure $Q_3 := \frac{1}{2}Q_1 + \frac{1}{2}Q_3$. By strict convexity we obtain

$$L(Q_3) < \frac{1}{2}L(Q_1) + \frac{1}{2}L(Q_3) = L^*, \tag{25}$$

which is a contradiction to $Q_1$ and $Q_2$ being global minimisers. $\qquad\square$

Note that in the literature on GVI (Knoblauch et al., 2022) it is common to assume that the regulariser is definite, i.e.

$$D(P, Q) = 0 \iff P = Q \tag{26}$$

for all $P, Q \in \mathcal{P}(\mathbb{R}^J)$. We did not use this assumption in neither Theorem 3 nor Theorem 4. However, the next lemma shows that it is basically implied by strict convexity.

---

[4]A sequence of probability measures $(Q_n)$ is called tight if and only if for every $\epsilon > 0$ there exists a compact set $K \in \mathbb{R}^J$ such that for all $n \in \mathbb{N}$ holds: $Q_n(K) > 1 - \epsilon$.

**Lemma 1.** *Let $D_P : \mathcal{P}(\mathbb{R}^J) \rightarrow [0, \infty]$ be strictly convex and assume further $D(Q, Q) = 0$ for all $Q \in \mathcal{P}(\mathbb{R}^J)$. Then it follows that $D(Q, P) = 0$ implies $P = Q$.*

*Proof.* We prove the claim by contradiction. Assume that there exists $P \neq Q$ such that $D(P, Q) = 0$. The strict convexity and $D(P, P) = 0$ imply combined that

$$D(\frac{1}{2}P + \frac{1}{2}Q, P) < \frac{1}{2}D(P, P) + \frac{1}{2}D(Q, P) \tag{27}$$
$$= 0. \tag{28}$$

However, we know that $D(\frac{1}{2}P + \frac{1}{2}Q, P) \geq 0$ by assumption. This is a contradiction. $\square$

**Discussion on loss assumptions** The assumptions on the loss $\ell$ in (L1) and (L3) are rather weak. Typically loss functions in machine learning are bounded from below and continuous (and therefore in particular lower semi-continuous). However, norm-coercivity can be violated. Consider for example the squared loss

$$\ell(\theta) := \sum_{n=1}^{N} \left(y_n - f_\theta(x_n)\right)^2, \tag{29}$$

where $f_\theta$ is the parametrisation of a neural network with one hidden layer, i.e. $\theta = (w, A)$ and

$$f_\theta(x) = w^T \sigma(Ax), \tag{30}$$

where $\sigma : \mathbb{R} \rightarrow \mathbb{R}$ is an activation function which is applied pointwise to the vector $Ax$ and has the property that $\sigma(0) = 0$. It is now possible to find a sequence of parameters $(\theta_k)_{k \in \mathbb{N}} \subset \mathbb{R}^J$ with $\|\theta_k\| \rightarrow \infty$ such that $\ell(\theta_k)$ does not converge to infinity. Define $w_k := k(1 \ldots 1)$, $A_k := 0$ and $\theta_k = (w_k \ A_k)$ for $k \in \mathbb{N}$. Then we obviously have that

$$\|\theta_k\| = \|w_k\| \rightarrow \infty \tag{31}$$

for $k \rightarrow \infty$ but

$$\ell(\theta_k) = \sum_{n=1}^{N} \left(y_n - f_{\theta_k}(x_n)\right)^2 \tag{32}$$

$$= \sum_{n=1}^{N} \left(y_n - w^T \sigma(0)\right)^2 \tag{33}$$

$$= \sum_{n=1}^{N} y_n^2, \tag{34}$$

which is constant and therefore does not converge to $\infty$. A similar, but notationally more involved, construction can be made for neural networks with more than one hidden layer. However, this is an issue that can be easily resolved by adding what is known as weight decay to the loss. For example, consider for $\gamma > 0$ the loss

$$\ell(\theta) := \sum_{n=1}^{N} \left(y_n - f_\theta(x_n)\right)^2 + \gamma \|\theta\|^2 \tag{35}$$

with weight decay. This loss is by construction norm-coercive and therefore the previous existence proof applies.

**Discussion on regulariser assumptions** The assumptions (D1) and (D2) are quite weak. The KL-divergence for example is known to be lower semi-continuous (Polyanskiy and Wu, 2014, Theorem 3.7) and strictly convex (Polyanskiy and Wu, 2014, Theorem 4.1). This immediately implies lower semi-continuity and convexity of $\mathrm{KL}(\cdot, P)$ for any fixed $P$. The MMD is also known to be strictly convex (Arbel et al., 2019, Lemma 25), whenever it is well-defined, which can be guaranteed under weak assumptions on $\kappa$ (Muandet et al., 2017, Lemma 3.1). The lower semi-continuity properties also depend on the kernel $\kappa$. However, for bounded kernels it is trivial to verify. We include the proof for completeness, but assume this has been shown before elsewhere.

**Lemma 2.** *Let the kernel $\kappa : \mathbb{R}^J \times \mathbb{R}^J$ be continuous and bounded: $\|\kappa\|_\infty := \sup_{\theta, \theta' \in \mathbb{R}^J} |k(\theta, \theta')| < \infty$ and $P$ be fixed. Then $\mathrm{MMD}(\cdot, P)$ is continuous and therefore, in particular, lower semi-continuous.*

*Proof.* Let $(Q_n)_{n \in \mathbb{N}}$ and $Q^*$ be such that

$$Q_n \xrightarrow{\mathcal{D}} Q^* \tag{36}$$

for $n \to \infty$. This immediately implies that

$$Q_n \otimes Q_n \xrightarrow{\mathcal{D}} Q^* \otimes Q^* \tag{37}$$

for $n \to \infty$, where $Q^* \otimes Q^*$ denotes the product measure of $Q^*$ with itself. Further, note that the kernel mean embedding $\mu_P$ is continuous as integral with respect to the second component of a continuous function and bounded since

$$|\mu_P(\theta)| = |\int \kappa(\theta, \theta') \, dP(\theta')| \tag{38}$$

$$\leq \int |\kappa(\theta, \theta')| dP(\theta') \tag{39}$$

$$\leq \|\kappa\|_\infty. \tag{40}$$

By the definition of weak convergence for measures, we therefore have

$$\iint \kappa(\theta, \theta') \, d(Q_n \otimes Q_n)(\theta, \theta') \longrightarrow \iint \kappa(\theta, \theta') \, d(Q^* \otimes Q^*)(\theta, \theta') \tag{41}$$

$$\int \mu_P(\theta) \, dQ_n(\theta) \longrightarrow \int \mu_P(\theta) \, dQ^*(\theta) \tag{42}$$

for $n \to \infty$. This immediately implies continuity of $\mathrm{MMD}(\cdot, P)$ with respect to the topology of weak convergence. $\square$

Notice that most kernels common in machine learning, such as the squared exponential or the Matérn kernel, are continuous and bounded and therefore Lemma 2 applies.

**Remark 1.** The astute reader may have noticed that our existence proof only guarantees the existence of measure $Q^* \in \mathcal{P}(\mathbb{R}^J)$. However, the Wasserstein gradient flow is by definition only formulated in the space of probability measures with finite second moment, denoted $\mathcal{P}_2(\mathbb{R}^J)$. Assumptions which guarantee that $Q^* \in \mathcal{P}_2(\mathbb{R}^J)$ are easy to formulate. For example, we can require that there exists $C > 0$ and $R > 0$ such that the loss $\ell$ satisfies

$$|\ell(\theta)| > C\|\theta\|^2 \tag{43}$$

for all $\|\theta\| > R$. This immediately implies that $Q^* \in \mathcal{P}_2(\mathbb{R}^J)$ since otherwise

$$\int |\ell(\theta)| \, dQ^*(\theta) = \infty \tag{44}$$

gives a contradiction to the finiteness of $L(Q^*)$. However, even if (43) is violated, the reference measure $P$ may still guarantee that $Q^* \in \mathcal{P}_2(\mathbb{R}^J)$. For example, if $P \in \mathcal{P}_2(\mathbb{R}^J)$, then $D_P(Q^*)$ will typically be large if $Q^* \notin \mathcal{P}_2(\mathbb{R}^J)$ and the global minimiser is therefore in a sense *unlikely* to have fat tails. We therefore assume $Q^* \in \mathcal{P}_2(\mathbb{R}^J)$ throughout the paper and consider it to be a minor practical concern.

## B   Realising the Wasserstein gradient flow

In this section, we identify a suitable stochastic process that allows us to follow the WGF.

Let $L^{\mathrm{fe}} : \mathcal{P}(\mathbb{R}^J) \to (-\infty, \infty]$ be the free energy discussed in Section 3.2 given as

$$L^{\mathrm{fe}}(Q) := \int V(\theta) \, dQ(\theta) + \frac{\lambda_1}{2} \int \kappa(\theta, \theta') \, dQ(\theta) dQ(\theta') + \lambda_2 \int \log \big(q(\theta)\big) q(\theta) \, d\theta, \tag{45}$$

where $\lambda_1, \lambda_2 \geq 0$ are constants, $V : \mathbb{R}^J \to \mathbb{R}$ is the potential, $\kappa : \mathbb{R}^J \times \mathbb{R}^J \to \mathbb{R}$ is symmetric. We will write $L$ for $L^{\text{fe}}$ from now on to simplify notation. The Wasserstein gradient of $L$ is given as (cf. Chapter 9.1 Villani, 2003, Equation 9.4)

$$\nabla_W L[Q](\theta) = \nabla V(\theta) + \lambda_1 (\nabla_1 \kappa * Q)(\theta) + \lambda_2 \nabla \log(q(\theta)), \tag{46}$$

where $\nabla_1 \kappa : \mathbb{R}^J \times \mathbb{R}^J \to \mathbb{R}^J$ is the (vector-valued) derivative of $\kappa$ with respect to the first component, $\nabla$ denotes the euclidean gradient with respect to $\theta$ and $(\nabla_1 \kappa * Q)(\theta) := \int \nabla_1 \kappa(\theta, \theta') \, dQ(\theta')$ for $\theta \in \mathbb{R}^J$. The corresponding Wasserstein gradient flow is therefore given as (cf. Chapter 9.1 Villani, 2003, Equation 9.3)

$$\partial_t q(t, \theta) = \nabla \cdot \Big( q(t, \theta) \big( \nabla V(\theta) + \lambda_1 (\nabla_1 \kappa * Q)(\theta) + \lambda_2 \nabla \log(q_t(\theta)) \big) \Big). \tag{47}$$

In general the probability density evolution of a stochastic process is—via the Fokker-Planck equation—associated with the adjoint of the (infinitesimal) generator of the stochastic process. We will therefore try to identify the generator associated to the density evolution in (47). To this end let $h \in C_c^2(\mathbb{R}^J, \mathbb{R})$ where $C_c^2(\mathbb{R}^J, \mathbb{R})$ denotes the space of twice continuously differentiable functions with compact support. We multiply both sides of (47) with $h$, integrate, and apply the partial integration rule to obtain

$$\frac{d}{dt} \int h(\theta) q(t, \theta) \, d\theta = - \int \nabla_W L[Q(t)](\theta) \cdot \nabla h(\theta) \, q(t, \theta) \, d\theta. \tag{48}$$

$$= - \int \big( \nabla V(\theta) + \lambda_1 (\nabla_1 \kappa * Q_t)(\theta) \big) \cdot \nabla h(\theta) \, dQ_t(\theta) \tag{49}$$

$$- \lambda_2 \int \nabla \log(q_t(\theta)) \cdot \nabla h(\theta) \, dQ_t(\theta). \tag{50}$$

By chain-rule and partial integration, (50) can be rewritten as

$$-\lambda_2 \int \nabla \log(q_t(\theta)) \cdot \nabla h(\theta) \, dQ_t(\theta) = -\lambda_2 \int \nabla q_t(\theta) \cdot \nabla h(\theta) \, d\theta \tag{51}$$

$$= \lambda_2 \int \Delta h(\theta) \, dQ_t(\theta). \tag{52}$$

Putting everything together, we obtain

$$\frac{d}{dt} \int h(\theta) q(t, \theta) \, d\theta = \int \big( A[Q(t)]h \big)(\theta) \, dQ_t(\theta), \tag{53}$$

where $\big\{ A[Q] \big\}_{Q \in \mathcal{P}(\mathbb{R}^J)}$ is a family of operators defined as

$$\big( A[Q]h \big)(\theta) := - \Big( \nabla V(\theta) + \lambda_1 (\nabla_1 \kappa * Q)(\theta) \Big) \cdot \nabla h(\theta) + \lambda_2 \Delta h. \tag{54}$$

for $h \in C_c^2(\mathbb{R}^J, \mathbb{R})$. The reader may recognize this operator family as the generator of a so called *nonlinear Markov processes* (Kolokoltsov, 2010, Chapter 1.4). The nonlinearity in this case refers to the dependency on the measure $Q$. Linear Markov processes have no measure-dependency. This family of generators corresponds to a McKean-Vlasov process of the form

$$d\theta(t) = - \Big( \nabla V(\theta(t)) + \lambda_1 (\nabla_1 \kappa * Q_t)(\theta(t)) \Big) dt + \sqrt{2\lambda_2} \, dB(t), \tag{55}$$

where $\big( B(t) \big)_{t>0}$ is a Brownian motion and $Q_t$ the law of $\theta(t)$. In other words: The solution to (55) has the time marginals $Q(t)$ such that (53) holds for every $h \in C_c^2(\mathbb{R}^J, \mathbb{R})$. Furthermore, the corresponding pdfs $\big( q(t) \big)$ satisfy the nonlinear Fokker-Planck equation given as

$$\partial_t q_t = A^*[Q_t] q_t, \tag{56}$$

where $A^*[Q]$ denotes the $L^2$-adjoint of the operator $A[Q]$ and is given as

$$\big( A^*[Q]h \big)(\theta) = \nabla \cdot \Big( h(\theta) \big( \nabla V(\theta) + \lambda_1 (\nabla_1 \kappa * Q)(\theta) + \lambda_2 \nabla \log(h(\theta)) \big) \Big) \tag{57}$$

for $h \in C_c^2(\mathbb{R}^J, \mathbb{R})$(Barbu and Röckner, 2020, cf. equation (1.1)-(1.4)). Note that (56) corresponds exactly to the Wasserstein gradient flow equation in (47). We can therefore follow the WGF by simulating solutions to (55).

The standard approach to simulate solutions to (55) (Veretennikov, 2006) is to use an ensemble of interacting particles. Formally, we replace $Q(t)$ by $\frac{1}{N_E} \sum_{n=1}^{N_E} \delta_{\theta_n(t)}$ and obtain

$$d\theta_n(t) = -\Big(\nabla V\big(\theta_n(t)\big) + \frac{\lambda_1}{N_E} \sum_{j=1}^{N_E} (\nabla_1 \kappa)\big(\theta_n(t), \theta_j(t)\big)\Big)dt + \sqrt{2\lambda_2}dB_n(t) \qquad (58)$$

for $n = 1, \ldots, N_E$ where $N_E \in \mathbb{N}$ denotes the number of particles. The Euler-Maruyama approximation of (58) leads to the final algorithm:

**Step 1:** Initialise $N_E \in \mathbb{N}$ particles $\theta_{1,0}, \ldots, \theta_{N_E,0}$ from a use chosen initial distribution $Q_0$.

**Step 2:** Evolve the particles forward in time according to

$$\theta_{n,k+1} = \theta_{n,k} - \eta\Big(\nabla V\big(\theta_{n,k}\big) + \frac{\lambda_1}{N_E} \sum_{j=1}^{N_E} (\nabla_1 \kappa)\big(\theta_{n,k}, \theta_{j,k}\big)\Big) + \sqrt{2\eta\lambda_2}Z_{n,k} \qquad (59)$$

for $n = 1, \ldots, N_E$, $k = 0, \ldots, T-1$ with $Z_{n,k} \sim \mathcal{N}(0, I_{J \times J})$.

Note that $\theta_{n,k}$ is thought of as approximation of $\theta_n(t)$ at position $t = k\eta$. Furthermore, as discussed in Section 4, various choices of $V$, $\lambda_1$ and $\lambda_2$ allow us to implement the WGF for different regularised optimisation problems in the space of probability measures. This is summarised below:

- Deep ensembles: $V(\theta) = \ell(\theta)$, $\lambda_1 = 0$, $\lambda_2 = 0$
- Deep Langevin ensembles: $V(\theta) = \ell(\theta) - \lambda \log p(\theta)$, $\lambda_1 := 0$, $\lambda := \lambda_2$
- Deep repulsive Langevin ensembles: $V(\theta) = \ell(\theta) - \lambda_1 \log p(\theta) - \lambda_2 \mu_P(\theta)$

## C  Asymptotic distribution of particles: unregularised objective

In this section, we investigate the asymptotic distribution of the WGF for the objective

$$L(Q) := \int \ell(\theta)\, dQ(\theta) \qquad (60)$$

for $Q \in \mathcal{P}(\mathbb{R}^J)$. The associated particle method is:

- Sample $\theta_1(0), \ldots, \theta_{N_E}(0)$ independently from $Q_0$.
- Simulate (deterministically) $\theta_n'(t) = -\nabla\ell\big(\theta_n(t)\big)$ for $n = 1, \ldots, N_E$.

We start by introducing some notation for the deterministic gradient system. Let $\phi^t(\theta_0)$ denote the solution to the ordinary differential equation (ODE)

$$\theta(0) = \theta_0 \in \mathbb{R}^J \qquad (61)$$
$$\theta'(t) = -\nabla\ell\big(\theta(t)\big) \qquad (62)$$

at time $t > 0$. In a first step, we show the following lemma, which is a simple application of the famous Lojasiewicz theorem (Colding and Minicozzi II, 2014), and the fact that Lebesgue almost every initialisation leads to a local minimum (Lee et al., 2016).

**Lemma 3.** *Assume $\ell : \mathbb{R}^J \to \mathbb{R}$ is norm-coercive and satisfies the Lojasiewicz inequality, i.e. for every $\theta \in \mathbb{R}^J$ exists an environment $U$ of $\theta$ and constants $0 < \gamma < 1$ and $C > 0$ such that*

$$|\ell(\theta) - \ell(\bar{\theta})|^\gamma < C|\nabla\ell(\theta)|. \qquad (63)$$

*for all $\bar{\theta} \in U$. Then we know that $\phi^t(\theta_0)$ converges for $t \to \infty$ to a local minimum of $\ell$ for Lebesgue almost every $\theta_0 \in \mathbb{R}^J$.*

*Proof.* First we show that $t \mapsto \phi^t(\theta_0)$ is bounded. We proof this by contradiction. Assume that $\phi^t(\theta_0)$ is unbounded. Then there exists a subsequence $(t_n)_{n \in \mathbb{N}} \subset [0, \infty)$ with $t_n \to \infty$ for $n \to \infty$ such that

$$|\phi^{t_n}(\theta_0)| \to \infty \tag{64}$$

for $n \to \infty$. The norm-coercivity immediately implies that

$$\ell(\phi^{t_n}(\theta_0)) \to \infty \tag{65}$$

for $n \to \infty$. However, this contradicts

$$\ell(\phi^t(\theta_0)) \leq \ell(\phi^0(\theta_0)) = \ell(\theta_0) < \infty, \tag{66}$$

where the first inequality follows from the fact that $t \mapsto \ell(\phi^t(\theta_0))$ is decreasing, which is a consequence of

$$\frac{d}{dt}\ell(\phi^t(\theta_0)) = \nabla\ell(\phi^t(\theta_0))\frac{d}{dt}\phi^t(\theta_0) \tag{67}$$

$$= -|\nabla\ell(\phi^t(\theta_0))|^2 \leq 0. \tag{68}$$

Hence $t \mapsto \phi^t(\theta_0)$ is bounded. By the Bolzano-Weierstrass theorem we can find a sequence $(t_n)_{n \in \mathbb{N}} \subset [0, \infty)$ with $t_n \to \infty$ and a point $\theta_\infty \in \mathbb{R}^J$ such that

$$\phi^{t_n}(\theta_0) \to \theta_\infty \tag{69}$$

for $n \to \infty$. Hence $(\phi^t(\theta_0))_{t>0}$ has the accumulation point $\theta_\infty$. The Lojasiewicz theorem (Colding and Minicozzi II, 2014) allows us to deduce that

$$\phi^t(\theta_0) \to \theta_\infty \tag{70}$$

for $t \to \infty$, and that $\theta_\infty$ satisfies $\nabla\ell(\theta_\infty) = 0$.

It remains to show that $\theta_\infty$ is not a saddle point for Lebesgue almost every initial value $\theta_0$. However, this is very similar to the proof in Lee et al. (2016). The only difference is that one would need to use a continuous-time version of the stable manifold theorem, which is readily available, for example in Bressan (2003). $\qquad \square$

Let $\{m_i\}_{i \in \mathbb{N}}$ denote the local minima of $\ell$ which are by assumption countable. Denote further by

$$\Theta_i := \left\{\theta_0 \in \mathbb{R}^J : \lim_{t \to \infty} \phi^t(\theta_0) \to m_i\right\} \tag{71}$$

the domain of attraction for the minimum $m_i$. The next theorem is then an easy consequence of Lemma 3.

**Theorem 5.** *Assume that the loss function $\ell$ only has countably many local minima, is norm coercive, and satisfies the Lojasiewicz inequality. Let further $\theta_0 \sim Q_0$ for some $Q_0 \in \mathcal{P}(\mathbb{R}^J)$ such that $\sum_{i=1}^{\infty} Q_0(\Theta_i) = 1$. Then,*

$$\phi^t(\theta_0) \xrightarrow{\mathcal{D}} \sum_{i=1}^{\infty} Q_0(\Theta_i)\,\delta_{m_i} =: Q_\infty \tag{72}$$

*for $t \to \infty$. Here $\xrightarrow{\mathcal{D}}$ denotes convergence in distribution.*

*Proof.* Let $\theta_0 \in \mathbb{R}^J$ be fixed. Due to Lemma 3, we know that

$$\phi^t(\theta_0) \to \sum_{i=1}^{\infty} m_i \mathbb{1}\{\theta_0 \in \Theta_i\} \tag{73}$$

for Lebesgue almost every $\theta_0$ for $t \to \infty$. Here, $\mathbb{1}\{\cdot\}$ denotes the indicator function. Let $Y$ now be a random variable with law $Q_0$. By assumption, we know that $Y \in \Theta_i$ for some $i \in \mathbb{N}$ with probability 1. Hence,

$$\phi^t(Y) \to \sum_{i=1}^{\infty} m_i \mathbb{1}\{Y \in \Theta_i\} \tag{74}$$

almost surely for $t \to \infty$. Since almost sure convergence implies convergence in distribution, we conclude that

$$\phi^t(Y) \xrightarrow{\mathcal{D}} \mathcal{L}\Big(\sum_{i=1}^{\infty} m_i \mathbb{1}\{Y \in \Theta_i\}\Big), \tag{75}$$

where $\mathcal{L}(\cdot)$ denotes the law of a random variable. However, the law of the RHS is easily recognised as

$$\mathcal{L}\Big(\sum_{i=1}^{\infty} m_i \mathbb{1}\{Y \in \Theta_i\}\Big) = \sum_{i=1}^{\infty} Q_0(\Theta_i) \delta_{m_i}, \tag{76}$$

which concludes the proof. $\qquad\square$

**Remark 2.** Note that the condition

$$\sum_{i=1}^{\infty} Q_0(\Theta_i) = 1 \tag{77}$$

in Theorem 5 is easy to satisfy. According to Lemma 3 the set

$$\mathbb{R}^J \backslash \bigcup_{i=1}^{n} \Theta_i \tag{78}$$

has Lebesgue measure zero. Therefore, any $Q_0$ which has a density w.r.t. the Lebesgue measure will satisfy (77).

# D   Asymptotic distribution for deep Langevin ensembles

In this section, we analyse the objective

$$L(Q) := \int \ell(\theta) \, dQ(\theta) + \lambda \operatorname{KL}(Q, P) \tag{79}$$

for $Q \in \mathcal{P}(\mathbb{R}^J)$. The corresponding particle method is given as:

- Sample $\theta_1(0), \ldots, \theta_{N_E}(0)$ independently from $Q_0$.
- Simulate the SDE $d\theta_n(t) = -\nabla V\big(\theta_n(t)\big)dt + \sqrt{2\lambda}dB_n(t)$ for each $n = 1, \ldots, N_E$.

Recall that $V(\theta) = \ell(\theta) - \lambda \log p(\theta)$. This case is well-studied in the literature and known as Langevin diffusion. Under mild assumptions (Chiang et al., 1987; Roberts and Tweedie, 1996),

$$\theta_n(t) \xrightarrow{\mathcal{D}} Q_\infty \tag{80}$$

for $t \to \infty$ and each particle $n = 1, \ldots, N_E$ independently. The probability measure $Q_\infty$ has the density

$$q_\infty(\theta) = \frac{1}{Z} \exp\Big(-\frac{V(\theta)}{\lambda}\Big) \tag{81}$$

$$= \frac{1}{Z} \exp\Big(-\frac{\ell(\theta)}{\lambda}\Big)p(\theta), \tag{82}$$

where $Z > 0$ is the normalising constant. As a consequence, the WGF asymptotically produces samples from $Q_\infty$. However, it is a priori unclear that $Q_\infty$ is in fact the same as the global minimiser $Q^*$ of $L$.

We investigate this question by relating invariant measures to stationary points of the Wasserstein gradient.

**Definition 1.** (Liggett, 2010, Thm. 3.3.7) A measure $Q$ is called an invariant measure (for a given Feller-process) if

$$\int Ah(\theta) \, dQ(\theta) = 0 \tag{83}$$

for all $h \in C_c^2(\mathbb{R}^J)$. Here $A$ is the infinitesimal generator of the corresponding Feller-process.

Recall that the infinitesimal generator of the Langevin diffusion for $h \in C_c^2(\mathbb{R}^J)$ is given as

$$Ah = -\nabla V \cdot \nabla h + \lambda \Delta h. \tag{84}$$

**Definition 2.** A measure $Q \in \mathcal{P}_2(\mathbb{R}^J)$ is called a stationary point of the Wasserstein gradient if

$$\nabla_W L[Q](\theta) = 0 \tag{85}$$

for $Q-$almost every $\theta \in \mathbb{R}^J$.

In finite dimensions, it is well-known that a local minimiser is a stationary point of the gradient. This carries over to the infinite-dimensional case, with a similar proof. Since we could not find this result anywhere in the literature we included it for completeness.

**Lemma 4.** *Let $\widehat{Q}$ be a local minimiser of L, i.e. there exits and $\epsilon > 0$ such that*

$$L(\widehat{Q}) \leq L(Q) \tag{86}$$

*for all $Q$ with $W_2(\widehat{Q}, Q) \leq \epsilon$. Then $\widehat{Q}$ is a stationary point of the Wasserstein gradient in the sense of Definition 2.*

*Proof.* Let $h \in C_c^2(\mathbb{R}^J)$ be arbitrary and $\widehat{Q} \in \mathcal{P}_2(\mathbb{R}^J)$ be a local minimum of $L$. Further, let $\phi^t(\theta_0)$ be the solution to the initial value problem

$$\theta(0) = \theta_0 \tag{87}$$
$$\theta'(t) = \nabla h(\theta(t)) \tag{88}$$

for $t \in (-\epsilon, \epsilon)$ for some $\epsilon > 0$. We now define $Q(t) := \phi^t \# \widehat{Q}$ for $t \in (-\epsilon, \epsilon)$ where $f \# \mu$ denotes the push-forward of the measures $\mu$ through the function $f$. In the Riemannian interpretation of the Wasserstein space, $\left(Q(t)\right)_{t \in (-\epsilon, \epsilon)}$ is a curve in $\mathcal{P}_2(\mathbb{R}^J)$ with tangent vector $h$ at point $\widehat{Q}$ (Ambrosio et al., 2005, Chapter 8). We, further, define $f : (-\epsilon, \epsilon) \to \mathbb{R}$ as $f(t) := L(Q(t))$. Application of the chain-rule (Ambrosio et al., 2005, p. 233) gives

$$f'(0) = \frac{d}{dt} L(Q(t))\big|_{t=0} \tag{89}$$

$$= \langle \nabla_W L[Q(0)], \nabla h \rangle_{L^2(Q(0))} \tag{90}$$

$$= \int \nabla_W L[\widehat{Q}](\theta) \cdot \nabla h(\theta) \, d\widehat{Q}(\theta). \tag{91}$$

We know that $f$ has a local minimum at $t = 0$ and, therefore, $f'(0) = 0$ which gives

$$0 = \int \nabla_W L[\widehat{Q}](\theta) \cdot \nabla h(\theta) \, d\widehat{Q}(\theta). \tag{92}$$

Since (92) holds for arbitrary test functions $h \in C_c^2(\mathbb{R}^J)$ and as $C_c^2(\mathbb{R}^J)$ is dense in $L^2(\widehat{Q})$, we obtain that $\nabla_W L[\widehat{Q}](\theta) = 0$ for $\widehat{Q}$-a.e $\theta \in \mathbb{R}^J$. $\qquad \square$

The next lemma relates invariant measures and stationary points of the Wasserstein gradient for infinitesimal generators of the form (84). It will prove extremely useful to translate between the Langevin diffusion literature and our optimisation perspective.

**Lemma 5.** *Let $Q \in \mathcal{P}_2(\mathbb{R}^J)$ be such that $Q$ has a density $q$ with respect to the Lebesgue measure. Then, the following two statements are equivalent:*

- *$Q$ is a stationary point of the Wasserstein gradient.*

- *$Q$ is an invariant measure.*

*Proof.* Let $Q$ be a measure with density $q$. Recall that the generator of the Langevin diffusion is for $h \in C_c^2(\mathbb{R}^J)$ given as

$$Ah = -\nabla V \cdot \nabla h + \lambda \Delta h. \tag{93}$$

By partial integration, it is easy to verify that the $L^2$- adjoint (w.r.t the Lebesgue measure) is given as

$$A^*h = \nabla \cdot (h \cdot \nabla V) + \lambda \Delta h. \tag{94}$$

We, therefore, conclude that

$$\int Ah(\theta) \, dQ(\theta) = \int Ah(\theta)q(\theta) \, d\theta \tag{95}$$

$$= \int h(\theta)A^*q(\theta) \, d\theta \tag{96}$$

$$= \int h(\theta)\Big(\nabla \cdot (q(\theta) \cdot \nabla V(\theta)) + \lambda \Delta q(\theta)\Big) \, d\theta. \tag{97}$$

Furthermore, we have $\nabla_W L[Q] = \nabla V + \lambda \nabla \log q$, and therefore

$$\int \nabla_W L[Q](\theta) \cdot \nabla h(\theta) \, dQ(\theta) = \int \nabla_W L[Q](\theta) \cdot \nabla h(\theta)q(\theta) \, d\theta \tag{98}$$

$$= \int \Big(\nabla V(\theta)q(\theta) + \lambda \nabla q(\theta)\Big) \cdot \nabla h(\theta) \, d\theta \tag{99}$$

$$= -\int h(\theta)\Big(\nabla \cdot (q(\theta)\nabla V(\theta)) + \lambda \Delta q(\theta)\Big) \, d\theta, \tag{100}$$

where the last line follows from applying partial integration. This allows us to conclude that

$$\int Ah(\theta) \, dQ(\theta) = -\int \nabla_W L[Q](\theta) \cdot \nabla h(\theta) \, dQ(\theta) \tag{101}$$

whenever $Q$ has a density. As a consequence we have that $Q$ is invariant if and only if it is a stationary point of the Wasserstein gradient. $\qquad\square$

Lemma 5 allows us to move between the optimisation and stochastic differential equation perspective. In Appendix A, we discussed the existence and uniqueness of a global minimiser $Q^*$ of $L$. We know that $Q^*$ has a density since the Kullback-Leibler divergence would be infinite otherwise (assuming $P$ has a Lebesgue-density which we assume throughout the paper). Lemma 4 guarantees that $Q^*$ is a stationary point of the Wasserstein gradient. Due to Lemma 5, we can infer that $Q^*$ must be an invariant measure. However, due to the uniqueness of the invariant measure under the previously mentioned mild assumptions (Chiang et al., 1987; Roberts and Tweedie, 1996), we can conclude that $Q^* = Q_\infty$.

## E    Asymptotic distribution of deep repulsive Langevin ensembles

In this section, we consider

$$L(Q) = \int \ell(\theta) \, dQ(\theta) + \frac{\lambda_1}{2} \operatorname{MMD}(Q, P)^2 + \lambda_2 \operatorname{KL}(Q, P) \tag{102}$$

$$= \int V(\theta) \, dQ(\theta) + \frac{\lambda_1}{2} \int \kappa(\theta, \theta') \, dQ(\theta)dQ(\theta') - \lambda_2 H(Q) + const, \tag{103}$$

as optimisation objective. Here, $H(Q) = -\int \log q(\theta)q(\theta) \, d\theta$ denotes the differential entropy.

Recall that in this case $V(\theta) = \ell(\theta) - \lambda_1 \mu_P(\theta) - \lambda_2 \log p(\theta)$. We already discussed in Appendix B that the McKean-Vlasov process of the form

$$\theta(0) \sim Q_0 \tag{104}$$

$$d\theta(t) = -\Big(\nabla V(\theta(t)) + \lambda_1(\nabla_1 \kappa * Q_t)(\theta(t))\Big)dt + \sqrt{2\lambda_2}dB(t), \tag{105}$$

with $\big(B(t)\big)_{t \geq 0}$ being a Brownian motion achieves the desired density evolution. Furthermore, the particle approximation of (104) is given as

$$d\theta_n(t) = -\Big(\nabla V\big(\theta_n(t)\big) + \frac{\lambda_1}{N_E}\sum_{j=1}^{N_E}(\nabla_1 \kappa)\big(\theta_n(t), \theta_j(t)\big)\Big)dt + \sqrt{2\lambda_2}dB_n(t) \tag{106}$$

for $n = 1, \ldots, N_E$ where $N_E \in \mathbb{N}$ denotes the number of particles.

The approach follows the same procedure as in Appendix D. We show the notions of invariant measures and stationary points of the Wasserstein gradient are the same for measures with Lebesgue density. We start by introducing the concept of an invariant measure for a nonlinear Markov process (Ahmed and Ding, 1993, Definition 1).

**Definition 3.** A measure $Q$ is called an invariant measure for a nonlinear Markov process with the family of infinitesimal generators $\big\{ A[Q] \big\}_{Q \in \mathcal{P}(\mathbb{R}^J)}$ if

$$\int A[Q]h(\theta) \, dQ(\theta) = 0 \tag{107}$$

for all $h \in C_c^2(\mathbb{R}^J)$.

Recall that the family of infinitesimal generators in our case is given as

$$\big( A[Q]h \big)(\theta) := -\Big( \nabla V(\theta) + \lambda_1 (\nabla_1 \kappa * Q)(\theta) \Big) \cdot \nabla h(\theta) + \lambda_2 \Delta h. \tag{108}$$

for $h \in C_c^2(\mathbb{R}^J, \mathbb{R})$. In analogy to Lemma 5, we obtain the following result.

**Lemma 6.** Let $Q \in \mathcal{P}_2(\mathbb{R}^J)$ be such that $Q$ has a density $q$ with respect to the Lebesgue measure. Then, the following two statements are equivalent:

- $Q$ is a stationary point of the Wasserstein gradient for $L$ in (102) in the sense of Def. 2.

- $Q$ is an invariant measure for the McKean-Vlasov process with infinitesimal generator defined in (108)

*Proof.* First, we notice that

$$\int A[Q]h(\theta) dQ(\theta) = \int A[Q]h(\theta)q(\theta) \, d\theta \tag{109}$$

$$= \int h(\theta) \big( A^*[Q]q \big)(\theta) \, d\theta. \tag{110}$$

Recall, that $A^*[Q]$ denotes the $L^2$-adjoint of the operator $A[Q]$ and that it is given as

$$\big( A^*[Q]h \big)(\theta) = \nabla \cdot \Big( h(\theta) \big( \nabla V(\theta) + \lambda_1 (\nabla_1 \kappa * Q)(\theta) + \lambda_2 \nabla \log \big( h(\theta) \big) \big) \Big) \tag{111}$$

for $h \in C^2(\mathbb{R}^J, \mathbb{R})$ with compact support. This implies

$$\big( A^*[Q]q \big)(\theta) = \nabla \cdot \Big( q(\theta) \big( \nabla V(\theta) + \lambda_1 (\nabla_1 \kappa * Q)(\theta) \big) \Big) + \lambda_2 \Delta q(\theta). \tag{112}$$

We plug this into (110) to obtain

$$\int A[Q]h(\theta) \, dQ(\theta) \tag{113}$$

$$= \int h(\theta) \nabla \cdot \Big( q(\theta) \big( \nabla V(\theta) + \lambda_1 (\nabla_1 \kappa * Q)(\theta) \big) \Big) \, d\theta + \int \lambda_2 h(\theta) \Delta q(\theta) \, d\theta. \tag{114}$$

On the other hand, we have that

$$\nabla_W L[Q](\theta) = \nabla V(\theta) + \lambda_1 (\nabla_1 \kappa * Q)(\theta) + \lambda_2 \nabla \log q(\theta), \tag{115}$$

and therefore

$$\int \nabla L[Q](\theta) \cdot \nabla h(\theta) \, dQ(\theta) \tag{116}$$

$$= \int \Big( \nabla V(\theta) + \lambda_1 (\nabla_1 \kappa * Q)(\theta) + \lambda_2 \nabla \log q(\theta) \Big) \cdot \nabla h(\theta) \, dQ(\theta) \tag{117}$$

$$= \int q(\theta) \big( \nabla V(\theta) + \lambda_1 (\nabla_1 \kappa * Q)(\theta) \big) \cdot \nabla h(\theta) \, d\theta + \lambda_2 \int \nabla q(\theta) \cdot \nabla h(\theta) \, d\theta \tag{118}$$

$$= -\int \nabla \cdot \Big( q(\theta) \big( \nabla V(\theta) + \lambda_1 (\nabla_1 \kappa * Q)(\theta) \big) \Big) h(\theta) \, d\theta - \lambda_2 \int q(\theta) \Delta h(\theta) \, d\theta, \tag{119}$$

where the last line follows from partial integration. Comparing (114) to (119) gives

$$\int A[Q]h(\theta)\, dQ(\theta) = -\int \nabla L[Q](\theta) \cdot \nabla h(\theta)\, dQ(\theta) \tag{120}$$

for all $h \in C_c^2(\mathbb{R}^J)$ whenever $Q$ has a density. This immediately implies that $Q$ is invariant iff it is a stationary point. $\qquad\square$

Again, we leverage this correspondence between stationary point and invariant measures. There is a rich literature on ergodicity of nonlinear Markov processes. For example, Theorem 2 of Veretennikov (2006) specifies conditions on $\kappa$ and $V$ such that

$$Q^{n,N_E}(t) \xrightarrow{\mathcal{D}} Q_\infty \tag{121}$$

for $N_E, t \to \infty$. Here $Q^{n,N_E}(t)$ denotes the law of a fixed particle $\theta_n(t)$, $n = 1, \ldots, N_E$, whose distribution is characterised by the SDE (58). The measure $Q_\infty$ is the unique invariant measure of the nonlinear Markov process. By Lemma 6 every invariant measure is a stationary point of the Wasserstein gradient and vice versa. Hence, existence and uniqueness of the stationary point of the Wasserstein gradient is immediately implied. However, since the global minimiser $Q^*$ is a stationary point of the Wasserstein gradient (cf. Lemma 4), we conclude by uniqueness that $Q_\infty = Q^*$.

# F  Asymptotic analysis of deep repulsive ensembles

In this section, we consider the objective

$$L(Q) := \int \ell(\theta)\, dQ(\theta) + \lambda \operatorname{MMD}(Q, P) \tag{122}$$

for $Q \in \mathcal{P}(\mathbb{R}^J)$. The corresponding McKean-Vlasov process is of the form

$$d\theta(t) = -\Big(\nabla V(\theta(t)) + \lambda(\nabla_1 \kappa * Q_t)(\theta(t))\Big) dt, \tag{123}$$

where $Q_t$ denotes the distribution of $\theta(t)$ and $V(\theta) = \ell(\theta) - \mu_P(\theta)$ with $\mu_P(\theta) = \int \kappa(\theta, \theta')\, dP(\theta)$ the kernel mean-embedding of $P$. We call the particle method in this case **deep repulsive ensembles (DRE)**.

The existence of the global minimiser $Q^*$ is still guaranteed under the assumptions in Appendix A. Lemma 4 guarantees that $Q^*$ is a stationary point of the Wasserstein gradient, i.e.

$$\nabla V(\theta) + \lambda(\nabla_1 \kappa * Q^*)(\theta) = 0 \tag{124}$$

for $Q^*$-a.e. $\theta \in \mathbb{R}^J$. Recall that the infinitesimal generator in this case is given as

$$\big(A[Q]h\big)(\theta) := -\Big(\nabla V(\theta) + \lambda(\nabla_1 \kappa * Q)(\theta)\Big) \cdot \nabla h(\theta) \tag{125}$$

for $Q \in \mathcal{P}(\mathbb{R}^J)$, $h \in C_c^2(\mathbb{R}^J)$. It immediately follows from the definition that

$$\big(A[Q]h\big)(\theta) = -\nabla_W L[Q](\theta) \cdot \nabla h(\theta) \tag{126}$$

for all $h \in C_c^2(\mathbb{R}^J)$, $\theta \in \mathbb{R}^J$. As in Lemma 5 & 6, this implies that each stationary point of the Wasserstein gradient is an invariant measure of the McKean-Vlasov process and vice versa. In Appendix D & E, we cite relevant literature that guarantees uniqueness of the invariant measure, which is a necessary (but not sufficient) condition for convergence to the invariant measure. The next theorem shows that uniqueness will in general not hold without the presence of the diffusion term.

**Theorem 6.** *The invariant measure for the McKean-Vlasov process with the family of generators $\big(A[Q]\big)_{Q \in \mathcal{P}(\mathbb{R}^J)}$ defined in (126) is (in general) not unique.*

*Proof.* Let $N_E \in \mathbb{N}$ and define $\widetilde{L} : \big(\mathbb{R}^J\big)^{N_E} \to \mathbb{R}$ as

$$\widetilde{L}(\theta_1, \ldots, \theta_{N_E}) := \sum_{i=1}^{N_E} V(\theta_i) + \frac{\lambda}{2N_E} \sum_{i,j=1}^{N_E} \kappa(\theta_i, \theta_j). \tag{127}$$

Assume that $V$ is bounded from below and norm-coercive. Then $\widetilde{L}$ is bounded from below and norm-coercive and therefore we can find a global minimiser $\theta^* := \left(\theta_1^*, \ldots, \theta_{N_E}^*\right) \in \left(\mathbb{R}^J\right)^{N_E}$ of $\widetilde{L}$. Since $\widetilde{L}$ is differentiable, we know that $\theta^*$ is a stationary point of the gradient which implies

$$\nabla V(\theta_i^*) + \frac{\lambda}{N_E} \sum_{j=1}^{N_E} (\nabla_1 \kappa)(\theta_i^*, \theta_j^*) = 0 \tag{128}$$

for all $i = 1, \ldots, N_E$. Here, we assume that the kernel $\kappa$ is symmetric, which is standard in the MMD literature. Note that (128) is equivalent to

$$\nabla V(\theta) + \lambda(\nabla_1 \kappa * \widehat{Q})(\theta) = 0 \tag{129}$$

for $\widehat{Q}$-a.e. $\theta \in \mathbb{R}^J$ where

$$\widehat{Q}(d\theta) := \frac{1}{N_E} \sum_{j=1}^{N_E} \delta_{\theta_j^*}(d\theta). \tag{130}$$

This means that $\widehat{Q}$ is a stationary point of the Wasserstein gradient, and therefore an invariant measure for the McKean-Vlasov process. Since $N_E \in \mathbb{N}$ was arbitrary, we have constructed countably many invariant measures and therefore uniqueness can't hold in general. $\qquad\square$

The reason that non-uniqueness of the invariant measure is an immediate contradiction to convergence is the following: If we initialise with any of the invariant measures constructed in the proof of Theorem 6, then the particle distribution of the McKean-Vlasov process will remain unchanged over time. Convergence to the global minimiser can therefore surely not hold for arbitrary initialisation $Q_0$. It may be possible to construct conditions on $Q_0$ under which convergence still holds. For example, for Stein variational gradient descent a similar issue occurs. However, in this case one can guarantee convergence (Lu et al., 2019, Theorem 2.8) if $Q_0$ has a Lebesgue-density (and if the kernel satisfies further restrictive assumptions). The existence of conditions that guarantee convergence for DRE remains an open problem.

## G   Implementation details

In Appendix A, we derived the following algorithm:

**Step 1:** Simulate $N_E \in \mathbb{N}$ particles $\theta_{1,0}, \ldots, \theta_{N_E,0}$ from a use chosen initial distribution $Q_0$.

**Step 2:** Evolve the particles forward in time according to

$$\theta_{n,k+1} = \theta_{n,k} - \eta\Big(\nabla V\big(\theta_{n,k}\big) + \frac{\lambda_1}{N_E} \sum_{j=1}^{N_E} (\nabla_1 \kappa)\big(\theta_{n,k}, \theta_{j,k}\big)\Big) + \sqrt{2\eta\lambda_2} Z_{n,k} \tag{131}$$

for $n = 1, \ldots, N_E$, $k = 0, \ldots, K-1$ with $Z_{n,k} \sim \mathcal{N}(0, I_{J \times J})$.

We can generate samples from DE, DLE and DRLE by setting the potential and regularisation parameters as described below:

- Deep ensembles: $V(\theta) = \ell(\theta)$, $\lambda_1 = 0$, $\lambda_2 = 0$
- Deep Langevin ensembles: $V(\theta) = \ell(\theta) - \lambda \log p(\theta)$, $\lambda_1 = 0$, $\lambda = \lambda_2$
- Deep repulsive Langevin ensembles: $V(\theta) = \ell(\theta) - \lambda_1 \log p(\theta) - \lambda_2 \mu_P(\theta)$

Due to Appendix D & E, we can think of $\theta_{1,K}, \ldots, \theta_{N_E,K}$ as approximately sampled from the global minimiser $Q^*$ for DLE and DRLE if $K$ is large enough. All experiments use the SE kernel given as

$$\kappa(\theta, \theta') = \exp\Big(-\frac{\|\theta - \theta'\|^2}{2\sigma_\kappa^2}\Big) \tag{132}$$

with lengthscale parameter $\sigma_\kappa > 0$. The kernel mean embedding $\mu_P$ can easily be approximated as

$$\mu_P(\theta) = \frac{1}{M} \sum_{i=1}^{M} \kappa(\theta, \theta_i), \ \theta \in \mathbb{R}^J, \tag{133}$$

where $\theta_1, \ldots, \theta_M \sim P$ independently. We chose $M = 20$.

## G.1 Toy example: global minimiser

We describe details regarding the experiments conducted to produce Figure 2 below.

We generate $N_E = 300$ particles and make the following choices:

- Loss: $\ell(\theta) := \frac{3}{2}(\frac{1}{4}\theta^4 + \frac{1}{3}\theta^3 - \theta^2) - \frac{3}{8}$
- Prior: $P \sim \mathcal{N}(0, 1)$ and therefore $\log p(\theta) = -\frac{1}{2}\theta^2$
- Initialisation: $Q_0 = P$
- Reg. parameter: $\lambda_{DLE} = 1$, $\lambda_{DRLE} = 1$, $\lambda'_{DRLE} = 1$
- Step size: $\eta = 10^{-4}$, Iterations: $K = 100,000$
- Kernel lengthscale, $\sigma_\kappa$, is chosen according to the median heuristic (Garreau et al., 2017) based on samples from the prior $P$

The loss is constructed such that we have a global minimum at $\theta = -2$, a turning point at $\theta = 0$, and a local minimum at $\theta = 1$.

- Deep ensembles: The optimal $Q^*$ is a Dirac measure located at the global minimiser $\theta = -2$. However, as we proved in Theorem 1, the WGF produce samples from

$$Q_\infty(d\theta) = \frac{1}{2}\delta_{-2}(d\theta) + \frac{1}{2}\delta_1(d\theta),\tag{134}$$

  as $(-\infty, 0)$ is the region of attraction for the global minimum and $(0, \infty)$ for the local minimum which both have probability 0.5 under $Q_0 = P = \mathcal{N}(0, 1)$. In particular, $Q_\infty \neq Q^*$ as expected.

- Deep Langevin ensembles: The optimal measure has the pdf

$$q^*(\theta) \propto \exp\left(-\frac{\ell(\theta)}{\lambda}\right)p(\theta)\tag{135}$$

  for $\theta \in \mathbb{R}$. As expected the WGF produces samples from $Q^*$.

- Deep repulsive Langevin ensembles: The optimal $q^*$ for deep repulsive ensembles is harder to determine. From the condition that $q^*$ is a stationary point of the Wasserstein gradient, we can derive that $u(\theta) := \log q^*(\theta)$ satisfies the integro-differential equation

$$u'(\theta) = -\frac{1}{\lambda_2}V'(\theta) - \frac{\lambda_1}{\lambda_2}\int (\nabla_1\kappa)(\theta, \theta')\exp(u(\theta'))\,d\theta'\tag{136}$$

  with some initial value $u(0) = u_0$. In principle, we could choose $u_0$ such that $q(\theta) := \exp(u(\theta))$ integrates to 1. However, since we do not know the appropriate initial condition a priori, we choose an arbitrary $u_0$ and normalise the pdf afterwards. We use an numerical solver to evaluate $u(\theta)$ on a fixed grid. As expected, the WGF produces samples from $Q^*$ in this case.

## G.2 Toy example: multimodal loss

The details below correspond to the experimental results presented in Figures 3 and 5. Figure 5 is an alteration of Figure 3 with only 4 particles with the goal of stressing the importance of the number of particles relative to the number of local minima.

**DE, DLE, DRLE** We generate $N_E = 300$ particles and make the following choices:

- Loss: $\ell(\theta) = -\log\sum_{i=1}^4 \frac{1}{4}\mathcal{N}(\theta; \mu_i, I_2)$, $\theta \in \mathbb{R}^2$, $\mu_i = (\pm 3, \pm 3)^T$, $i = 1, \ldots, 4$
- Prior: $P$ flat and therefore $\log p(\theta) = 0$
- Initialisation: $Q_0 \sim \mathcal{N}(0, I_2)$
- Reg. parameter: $\lambda_{DLE} = 0.2$, $\lambda_{DRLE} = 0.2$, $\lambda'_{DRLE} = 0.6$
- Step size: $\eta = 0.1$, Iterations: $K = 10,000$

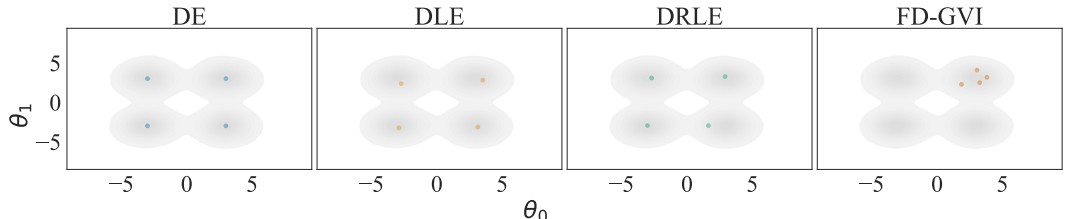

Figure 5: We generate $N_E = 4$ particles from DE, DLE, DRLE and FD-GVI with Gaussian parametrisation. The multimodal loss $\ell$ is plotted in grey and the particles of the different methods are layered on top. The prior in this example is flat, i.e. $\log p$ and $\mu_P$ are constant. The initialisation $Q_0$ is standard Gaussian.

- Kernel lengthscale, $\sigma_\kappa$, is chosen according to the median heuristic (Garreau et al., 2017) based on samples from the prior $P$

Note that for a translation-invariant kernel such as the SE kernel we obtain for the flat prior $P$ that

$$\mu_P(\theta) = \int_{-\infty}^{\infty} \kappa(\theta, \theta')\, d\theta' \tag{137}$$

$$= \int_{-\infty}^{\infty} \phi(\theta - \theta')\, d\theta' \tag{138}$$

$$= \int_{-\infty}^{\infty} \phi(\xi)\, d\xi, \tag{139}$$

where the second line follows from the fact that we can write any translation-invariant kernel as $\kappa(\theta, \theta') = \phi(\theta - \theta')$ for some function $\phi : \mathbb{R}^J \to \mathbb{R}$ and the second line is simple variable substitution. If (139) is finite, the above expression is well-defined and therefore $\mu_P$ constant. Note that in particular for the SE kernel, we have $\phi(\xi) = \exp(-\|\xi\|^2 / (2\sigma_\kappa^2))$ and therefore (139) is finite. As a consequence, we have that for a flat prior $P$ the gradient of the potential $V$ is the same for all three methods. This means that the loss $\ell$ isn't adjusted and the only difference between the three methods is the presence of repulsion and noise effects.

**Remark 3.** The astute reader may have noticed that a flat prior $P$ is in fact not covered by our theory in Appendix A. The problem is that $\mathrm{KL}(\cdot, \mathcal{L})$, where $\mathcal{L}$ denotes the Lebesgue measure, is not positive (and not even bounded from below). To see this, choose $Q = \mathcal{N}(0, \Sigma)$ with $\Sigma = \mathrm{diag}(\sigma_1^2, \sigma_2^2)$ and note that

$$\mathrm{KL}(Q, \mathcal{L}) = \int \log q(\theta)\, q(\theta)\, d\theta = -\mathrm{H}(Q), \tag{140}$$

where $\mathrm{H}(Q)$ denotes the differential entropy. For a Gaussian, it is known that

$$\mathrm{H}(Q) = \frac{1}{2} \log\left((2\pi e)^J \det(\Sigma)\right) = \frac{1}{2}\left(\log(2\pi e)^J + \log(\sigma_1^2) + \log(\sigma_2^2)\right) \tag{141}$$

and therefore if either $\sigma_1^2 \to \infty$ or $\sigma_2^2 \to \infty$ then $\mathrm{KL}(Q, \mathcal{L}) \to -\infty$. However, note that this difficulty is rather technical in nature and can easily be remedied. Instead of $\mathcal{L}$, we could have chosen the uniform prior $P \sim U(-10^{100}, 10^{100})$. In this case, the positivity of $\mathrm{KL}(\cdot, P)$ is guaranteed by Jensen's inequality. This choice of $P$ gives—up to an additive constant—the same objective as a flat prior and up to machine precision the same kernel mean embedding $\mu_P$. It is, therefore, algorithmically irrelevant if $P$ is flat or uniform on a very large set.

**FD-GVI** We use the same prior and loss as for DE, DLE and DRLE. We parameterise the variational family as independent Gaussian, i.e.

$$\mathcal{Q} = \left\{ \mathcal{N}(\mu, \Sigma) \,|\, \mu \in \mathbb{R}^2,\, \Sigma = \mathrm{diag}\left(\exp(\beta_1), \exp(\beta_2)\right), \beta := (\beta_1, \beta_2)^2 \in \mathbb{R}^2 \right\}. \tag{142}$$

We learn the variational parameters $\nu := (\mu, \beta) \in \mathbb{R}^4$ by minimising

$$\widetilde{L}(\nu) = \int \ell(\theta)\, dQ_\nu(\theta) + \lambda KL(Q_\nu, P) \tag{143}$$

$$= \int \ell(\theta)\, dQ_\nu(\theta) - \lambda H\big(\mathcal{N}(\mu, \Sigma)\big) \tag{144}$$

$$\approx \frac{1}{200} \sum_{j=1}^{200} \ell(\mu + \Sigma^{0.5} Z_j) - \frac{\lambda}{2} \log\big((2\pi e)^2 \exp(\beta_1) \exp(\beta_2)\big) \tag{145}$$

$$= \frac{1}{200} \sum_{j=1}^{200} \ell(\mu + \Sigma^{0.5} Z_j) - \frac{\lambda}{2}\big(\beta_1 + \beta_2\big) + const, \tag{146}$$

where $Z_1, \ldots, Z_{200} \sim \mathcal{N}(0, I_2)$ and $H\big(\mathcal{N}(\mu, \Sigma)\big)$ denotes the differential entropy of the normal distribution. For the regularisation parameter, we chose $\lambda = 0.5$.

### G.3 Toy example: more modes than particles

We generate $N_E = 20$ particles and make the following choices:

- Loss: $\ell(\theta) = -|\sin(\theta)|$, $\theta \in [-M\pi, M\pi]$, with $M = 1000$
- Prior: $P$ flat and therefore $\log p(\theta) = 0$ and $\mu_P = $ const. (cf. Appendix G.2)
- Initialisation: $Q_0 \sim U(-M\pi, M\pi)$
- Reg. parameter: $\lambda_{DLE} = 0.001$, $\lambda_{DRLE} = 0.001$, $\lambda'_{DRLE} = 0.6$
- Step size: $\eta = 0.01$, Iterations: $K = 1.000$
- Kernel lengthscale, $\sigma_\kappa$, is chosen according to the median heuristic (Garreau et al., 2017) based on samples from the prior $P$

Note that $\ell$ has $2M = 2000$ local minima at locations
$$m_i := \frac{\pi}{2} + i\pi, \quad i \in \{-M, \ldots, 0, \ldots, (M-1)\}. \tag{147}$$

Due to the flat prior $\nabla V = \nabla \ell$ for all three methods. We observe that it is hard to distinguish the methods since most particles are in their local modes by themselves.

### G.4 UCI Regression

The UCI data sets are licensed under Creative Commons Attribution 4.0 International license (CC BY 4.0). Following Lakshminarayanan et al. (2017), we train 5 one-hidden-layer neural networks $f_\theta$ with 50 hidden nodes for 40 epochs. We split each data set into train (81% of samples), validation (9% of samples), and test set (10% of samples). Based on the best hyperparameter runs (according to a Gaussian NLL) found via grid search on a validation data set, we make the following choices:

- Loss: $\ell(\theta) = \frac{1}{N} \sum_{n=1}^{N} (f_\theta(x_n) - y_n)^2$ where $\{x_n, y_n\}_{n=1}^{N}$ are paired observations.
- Prior: $P \sim \mathcal{N}(0, 1)$
- Initialisation: Kaiming intilisation, i.e. for each layer $l \in \{1, \ldots L\}$ that maps features with dimensionality $n_{l-1}$ into dimensionality $n_l$, we sample $Q_{l,0} \sim \mathcal{N}(0, 2/n_l)$
- Reg. parameter: $\lambda_{DLE} = 10^{-4}$, $\lambda_{DRLE} = 10^{-4}$, $\lambda'_{DRLE} = 10^{-2}$
- Step size: $\eta = 0.1$, Iterations: $K = 10,000$
- Kernel lengthscale, $\sigma_\kappa$, is chosen according to the median heuristic (Garreau et al., 2017) based on samples from the prior $P$

### G.5 Compute

While the final experimental results can be run within approximately an hour on a single GeForce RTX 3090 GPU, the complete compute needed for the final results, debugging runs, and sweeps amounts to around 9 days.

