# OpenReview forum: "A Rigorous Link between Deep Ensembles and (Variational) Bayesian Methods"
_NeurIPS.cc/2023/Conference — NeurIPS 2023 oral_

### Official Review · Reviewer_9edE · 2023-06-27

**Soundness:** 4 excellent
**Presentation:** 4 excellent
**Contribution:** 4 excellent
**Rating:** 8
**Confidence:** 2

**Summary:**

This paper provides a theoretical framework based on generalized variational inference [1] and Wasserstein gradient flows (WGF) for analyzing deep ensemble methods and their regularized versions. The authors demonstrate that deep ensembles and other variational Bayesian methods can be cast as instances of an infinite dimensional variational inference problem and the WGF of different instantiations of a free energy functional. The authors additionally use their theoretical framework to derive a new algorithm for generating samples from a target distribution.

[1] Knoblauch, Jeremias, Jack Jewson, and Theodoros Damoulas. "An optimization-centric view on Bayes’ rule: Reviewing and generalizing variational inference." Journal of Machine Learning Research 23.132 (2022): 1-109.


**Strengths:**

The paper is well-organized and written, and the benefits of the unifying theoretical framework are compelling. The discussion of how deep ensemble methods can be viewed through the lens of WGF and the use of this lens to prove theoretical guarantees on the limiting behavior of particle estimations is useful and insightful. This work also holds the promise of deriving new algorithms, as demonstrated by the deep repulsive Langevin ensembles presented in Section 4.3.


**Weaknesses:**

#### **Experiments section is difficult to follow**
While the details corresponding to the various figures in Section 5 are fully provided in Appendix G, this section is currently difficult to follow as a stand-alone section in the main paper. Without (even high level) details on the experimental setup and the general motivation of each experiment it is difficult to dive right into the results and Figures as they are currently presented. I recommend moving some details from Appendix G into the main text and providing the context for each experiment before diving into the results.

---

#### **Motivation for convexification is unclear**
While the authors prove that convexity of the infinite-dimensional variational form of the learning objective guarantees uniqueness of a minimizer, this is somewhat disconnected from the presented goal of optimizing $\ell(\theta)$ via probabilistic lifting. For example, in footnote 1 on page 2, the authors argue that the unregularized variational objective has non-unique optimum. However, the local optima all have equivalent values of the objective and are simply weighted averages of equivalent optima of $\ell$, hence it is not clear why uniqueness is a desiderata here.

Additionally, Figure 2 in Section 5 demonstrates how deep ensembles (DE) do not converge to $Q^*$. However, although deep Langevin ensembles (DLE) and deep repulsive Langevin ensembles (DRLE) provable converge $Q^*_{DLE}$ and $Q^*_{DRLE}$, respectively, these optimal distributions are also not equal to $Q^*$.

Hence a clearer exposition as to why regularized optima are preferred to the unregularized ones is needed.

---

#### **Motivation for DRLE is lacking**
While DRE / DRLE is indeed interesting as a new algorithm that can be derived from the presented theoretical framework, it would be great if the authors also provided some intuition / motivation as to why MMD is perhaps a better suited divergence regularizer than KL.


**Questions:**

In Lines 188-191, the authors state that
> In theory, the PDE in (4) provides us with a direct way of implementing infinite-dimensional gradient descent for (1): simply follow the WGF. In practice however, this is impossible: **numerical solutions to PDEs become computationally infeasible for the high-dimensional parameter spaces which are common in deep learning applications.**

I am unclear what is meant by this. Why is the high-dimensionality of deep learning parameterizations relevant here? Isn’t this problem simply impossible since it involves an infinite-dimensional space?


**Limitations:**

Authors can potentially elaborate on the future directions of this work, specifically around analyzing the approximation errors of approximating WGF with finite number of particles over a finite time horizon.

---

> ### Author Rebuttal · Authors · 2023-08-08
>
> #### Details on experimental section
>
> As the reviewer notes themselves, the paper is already rather densely packed. Because of space limitations, we were not able to include all relevant experimental details in the paper. We chose to focus on methodological details, but will move as many details into the main part in accordance with the available space in the final version.
>
> #### Motivation for convexification
>
> We thank the reviewer for their comment. One needs to distinguish between two questions. The first one relates to the uniqueness of the minimiser of the optimisation problem. The point that we try to make with the footnote is simple: without regularization, the optimization problem can in general not have a unique minimiser. This relates to the bigger question that the reviewer raises: Why should one care about uniqueness? The main problem with non-uniqueness is that this typically translates into undesirable properties of the inference algorithm—for example, unless the solution is unique, it is not even clear which solution (of the many possible ones) an algorithm should target. This is illustrated by Theorem 1: the asymptotic distribution of Deep Ensembles depends strongly on initializations and how large the domain of attraction for a given local minimum is. Therefore, if we initialize poorly, we may end up putting most particles close to a comparatively poor local minimum.
>
> Identifying a unique target to aim for, which in our case is $Q^*$, is typically seen as the first step in designing an inference algorithm with theoretical guarantees—after all, unless the target is clearly defined, it is not generally possible to assess an algorithm’s performance. Once the target has been identified, the second step is to build an inference algorithm that finds it.
>
> Regarding the interpretation of Figure 2: There may be some confusion about what $Q^*$ is in the plot. The dotted line in the background is the (unique and optimal) $Q^*$ of the optimisation problem; and the histogram clearly shows that DLE and DRLE do generate samples from this global minimiser (in line with our theory). On the other hand, DE does not (!). While it would be possible to find an initialisation distribution $Q_0$ such that DE converges to its global minimum, we would have to know where the global minimum (and its region of attraction) are located to do this—and this is precisely the kind of dependence on initialisation that is undesirable! DLRE and DLE converge to their respective optimal $Q^*$ regardless of initialization, thereby losing this undesirable trait. Importantly, note that this would be impossible to achieve if we hadn’t guaranteed the existence of a unique minimizer through convexification.
>
> #### Motivation for DRLE
>
> We thank the reviewer for this great question. It is a point that we will discuss more thoroughly in the final version of the manuscript. The MMD is based on the kernel $\kappa$ which introduces a repulsive effect between different sets of particles. This means that we can essentially choose our own metric that compares different parameter vectors (i.e. particles) with each other. It now depends heavily on the application, which sets of parameters should be considered ‘similar’. We chose a very basic kernel, the squared exponential, which essentially just measures the euclidean distance between the parameter vectors. However, in certain applications one might have a very good understanding of the type of diversity one would want to encourage and this knowledge could be embedded in the kernel $\kappa$. We considered a thorough investigation of the effects of the kernel to be beyond the scope of this paper, but it is an interesting avenue for future research.
>
>
> #### Questions
> [PDE solvers] We are grateful the reviewer pointed out that this sentence is misleading. We mean the following: Once we have a closed form expression for the Wasserstein gradient (as for example in the KL and MMD case) the PDE in (4) gives us a way to determine the pdf of $Q(t)$ at any time $t$. We can simply apply numerical PDE solvers to find an approximation of $q(t,\theta)$. However, PDE solvers quickly become computationally infeasible if the input space is larger than 2 or 3. As the parameter space in deep learning is typically at least in the thousands and more often than not in the millions or billions, it is computationally infeasible to deploy numerical PDE solvers.
>
> #### Limitations
> This is an excellent point. We have written a response to this in the general rebuttal point 3.

---

> > ### Comment · Reviewer_9edE · 2023-08-11
> > **Thank you**
> >
> > Thank you for the thorough response. I do not have any additional comments or questions at this time.

---

### Official Review · Reviewer_4EE1 · 2023-07-05

**Soundness:** 4 excellent
**Presentation:** 3 good
**Contribution:** 4 excellent
**Rating:** 8
**Confidence:** 2

**Summary:**

The paper established theoretical connections between ensembling an old and established method of deriving uncertainty estimates They use theory from iteraction of particles  in a thermodynamic system to generalise and connect seemingly different ways of ensembling and Variational Bayes(Inference) methods. This is done by formulating the original non-convex optimization problem ubiqitious in ML and stats as infinite dimensional convex optimization problem in the space of probability measures. The addition of a regularization quantity ensures the strict convexity of the problem and by choosing different forms of this quantity lead to derivation of various inference algorithms.

**Strengths:**

1. The paper is well written, theory heavy and addresses important topic of deep ensembling and its connection with variational Bayes methods
2. I am not so good with theory, but the theorems and equations looked ok to me without obvious mistakes.
3. Although this is a theory paper, the theoretical claims are well supported by the experiments and where they are not the authors they explain it well.
4. The distinction between IDGVI and FDGVI is well drawn out and explained. Also the limitations with FDGVI that the approximation family is limited by construction serves as a motivation for using IDGVI methods.

**Weaknesses:**

1. There is a lot of content that has been compressed in 9 pages which can be challenging for a reader, and a journal might have been more appropriate for this work.

**Questions:**

1. in practice, deep networks use stochastic gradients, does the theory hold for stochastic gradients, esp. as Section 2 explicitly uses gradients for motivation.
2. For case of FDGVI, the authors do not consider normalizing flows or SIVI as methods to overcome the limited capacity of approximating family problem when comparing it with IDGVI.


**Limitations:**

The limitations or practical challenges with the derived inference algorithms can be addressed. How practical is the result from Theorem 2, is it something that will only work asymptotically or will this work practically and if so how fast ?

---

> ### Author Rebuttal · Authors · 2023-08-08
>
> #### Questions
> 1. This is indeed an excellent question. It is true that in practice, we will need to replace full gradients by mini-batch versions and for example the kernel mean embedding with its monte carlo estimator. The reviewer correctly observes that the theoretical results in Section 4 do not account for this type of approximation and sub-sampling. However, this is not unique to our analysis, and a rather common simplification. More importantly, we believe that the results built by ignoring this added complication already provide the most important insights into the differences between various algorithms and can be used as a starting point for further theoretical investigations.
> More precisely, we think of the Wasserstein gradient flow as a powerful and principled tool to derive new inference algorithms for different types of regulariseres (MMD/KL and maybe others in the future). We can then combine it with standard plug-in estimators (like mini-batch gradient estimators) to immediately obtain an algorithm that approximately performs gradient descent in infinite dimensions.
> 2. We thank the reviewer for pointing us to these alternatives for VI. We have included references for VI via normalizing flows and SIVI  in the main text. To the more general point: It is indeed true that one can make the class of variational inference more expressive, but this comes with a trade-off: If we use a more expressive approximating family, the KL-divergence term is typically not available in closed form anymore, which means we need to introduce approximations for the regulariser. Furthermore, the resulting optimization problems for the variational parameters are still highly non-convex and therefore often depend heavily on good initialisations. We are not aware of any works in the FD-GVI literature that obtain results competitive with deep ensembles. We believe that this is a direct consequence of the above problems.
>
> #### Limitations
>
> The reviewer raises an excellent point. Results such as the one presented in Theorem 2 are only asymptotic in nature, and consequently we have no guarantee that the convergence is fast enough. These results should therefore be seen as necessary but not sufficient conditions for a good inference algorithm. Yet, stronger results which quantify the speed of convergence, would require us to make extremely strong assumptions that will be violated for deep learning. For example, we could trivially obtain quantitative results by citing the relevant literature albeit under very strong assumptions that would never be satisfied in the context of deep learning. Section 11.2 of  Ambrosio et al. (2005) shows that if $L$ is lambda-convex along generalized geodesics, we obtain exponentially fast convergence. Lambda-convexity of $L$ can be guaranteed if the potential $V$ is strictly convex (cf. Section 9.3 of Ambrosio et al. (2005)), which is surely never satisfied in deep learning applications.
>
> The final version will make this difference between qualitative and quantitative results clearer, and discuss the strong assumptions required to obtain quantitative results.
>
> However, we want to point out that standard parameterized FD-GVI comes with no guarantees of any type. Although qualitative results have their shortcomings, they at least show that the inference algorithm in principle is powerful enough to solve the optimization problem at hand.
>
>
>
> #### References
>
> Ambrosio, L., Gigli, N., and Savare, G. (2005). Gradient flows: in metric spaces and in the space of probability measures. Springer Science & Business Media.

---

> > ### Comment · Reviewer_4EE1 · 2023-08-14
> > **Response to the rebuttal**
> >
> > Thanks to the authors for replying to my questions. I am quite satisfied with their detailed comments to my questions and other reviewers' questions. I recommend a strong acceptance.

---

### Official Review · Reviewer_AtWA · 2023-07-06

**Soundness:** 4 excellent
**Presentation:** 4 excellent
**Contribution:** 4 excellent
**Rating:** 8
**Confidence:** 3

**Summary:**

The authors propose to unify existing theory on Bayesian (variational) inference (VI) by addressing a generalized objective, which is obtained from standard parameterized loss minimization by “probabilistic lifting” (re-casting in a space of probability measures over the parameter) and “convexification” (ensuring the existence of a global minimizer by regularization), with infinite-dimensional gradient flows in 2-Wasserstein space. A general recipe is provided to implement such a Wasserstein gradient flow (WGF) via an energy objective and a system of interacting particles. In the key contribution of the paper, the authors study WGF with different types of regularization–most notably, the unregularized version corresponding to deep ensembles (DE). It is shown that DE do not conduct a Bayesian learning procedure and systematically fail to generate samples from the optimal distribution, yet perform competitively thanks to the flexibility of the infinite-dimensional inference they realize (as opposed to, e.g., classical parametric VI).

**Strengths:**

* [S1] **Unifying framework**. After much discussion in the past few years, the authors are–to the best of my knowledge–the first to establish a comprehensive theory that encompasses (finite-dimensional) VI and DE.
* [S2] **Clarity**. Despite the rather abstract subject, the authors present a coherent and easy-to-follow sequence of arguments. Complexity is strictly limited to the necessary extent.
* [S3] **Rigor**. Mathematical concepts and notation are sound. Extensive proofs and/or references to prior work underline every claim (though I did not check every proof in detail).


**Weaknesses:**

* [W1] **Analysis of DE behavior** (see Questions)
  * It is not entirely clear if the paper studies arbitrary variants of DE or only a very narrowly defined version (see Q5).
  * The authors conjecture that the number of samples being vastly smaller than the number of local minima is responsible for D(R)LE not outperforming DE consistently and point to Fig. 4. This evidence seems rather anecdotal and could benefit from a more detailed investigation. Also see Q7--Q8.
* [W2] **Omissions in notation**. While the notation is consistent and comprehensible overall, the authors tend to omit integration domains, objects of differentiation etc. (e.g., Eq. 1, l. 150, l. 177, l. 179). With the shifting of integration spaces and various gradients involved, it would be helpful to be as explicit as possible in this regard.


**Questions:**

* [Q1] Eq. 6: What does the index $j$ relate to?
* [Q2] l. 218: Are there any convergence results in the respective limits of $T$ and $N_E$?
* [Q3] l. 223: The experiments are promised to confirm small approximation errors due to finite samples and time, in particular in comparison to finite-dimensional methods. Where, exactly, do I find evidence for this claim?
* [Q4] l. 236: Just for the sake of clarity, is $\theta^\prime_n(t) = - \nabla \ell(\theta_n(t))$ equivalent to $d \theta_n(t) - \nabla V(\theta_n(t)) dt$?
* [Q5] l. 237: Do I understand correctly that you interpret DE as training with no regularization whatsoever (weight decay, batch normalization etc. – let alone variations with weight sharing and the like)? I doubt that many researchers actually apply such a decidedly naive approach.
* [Q6] l. 253: Can DE implementing infinite-dimensional GD be understood as taking a non-parametric/functional approach as to what the distribution of the generated samples looks like (as opposed to, e.g., mean-field VI with a finite parameter vector)?
* [Q7] Fig. 2: Is there an explanation why DE exhibits this precise 50/50 spread of the probability mass?
* [Q8] Fig. 4: I’m not sure I understand the key message here. Why do the particles end up in the same modes despite different $Q^\ast$? What would be the expected behavior?

—

Minor remarks

* l. 53: Redundant “i” in “probability”
* l. 72: Shouldn’t $D$ provide the mapping $(Q, P) \mapsto D(Q, P)$ according to the stated domain?
* l. 170: I feel enough space can be freed to include the definition of the 2-Wasserstein metric, it seems an odd choice to omit this arguably relevant information.
* l. 292: Is there a $d$ missing in front of $Q(\theta)$ in the first double integral?
* l. 297: $L^{FE}$ with capitalized superscript ($L^{fe}$ otherwise).
* l. 329: White space after “minimisers”
* l. 330: Remove either “which” or “that”
* l. 337: “matters”
* Fig. 3 (caption): White space in “FD-GVI”
* Table 1: I would recommend removing the “Boston” dataset due to its racism issues. Also: “methods… outperform” or “method… outperforms”.
* l. 355: “lens”

**Limitations:**

Given that the main contribution is a unifying framework for existing theories, this point doesn’t apply as usual. However, the authors should state more clearly that the evidence shown in Section 5 for findings in Section 4 is quite limited.

---

> ### Author Rebuttal · Authors · 2023-08-08
>
> #### Weaknesses
> 1.  One of the aims of this paper is to show that the naive strategy of train-and-repat can be understood as Wasserstein gradient flow of the probabilistic lifting of the loss function. Which strategies are covered depends on a case-by-case basis. Weight-decay for example is covered by our theory, as it just means that $\ell$ is given as MSE+$\lambda |\theta|^2$. Batch-normalization is also covered in the sense that an ensemble of neural-networks which implement batch-normalization can be understood through the same WGF-lense. However, we did not want to give the impression that DEs are a bad strategy (even in their most basic form). Quite the opposite: we stress throughout the paper that they can be interpreted as implementing a rather sophisticated version of gradient descent and that especially in the multi-modal loss-landscapes of deep learning it is hard to come up with a better strategy.
> 2.  Thank you for pointing this out, we will make a more concerted effort at explaining this in the updated manuscript. As our theory shows, a key difference between DE and D(R)LE is the form the solutions take: with infinitely many particles, D(R)LE produces a unique probability measure that has a continuous density (see Figure 2). In contrast, Theorem 1 shows that DEs produce probability measures that have atomic support: they have probability measure zero almost everywhere—except at a few individual points. In a sense, DEs are an extremely sparse representation of parameterisations of the neural network we care about. This is different for the measures produced by D(R)LEs: (with infinitely many particles) because they are densities, they try to assign probability mass to all regions of the parameter space. It is reasonable to believe that this would lead to a less sparse/better representation of parameterisations of the neural network we care about. Figure 2 supports this idea: in the setting where there are many more particles than minima, we can cover the parameter space of the neural network well—and in these settings, the purely atomic nature of DEs would generally be a drawback. To show this, we sampled the initial values for the parameters for the DE uniformly between -2.5 and 2.5. Since there are two minima in Figure 2 whose basins of attraction for gradient descent are around [-2.5, 0] and [2.5,0], a uniform initialization leads to the exact 50:50 spread.
> However, in practice, neural nets don’t look like Figure 2—they have lots and lots of minima, and we have very few particles to try and approximate the densities that look so clean and regular in Figure 2. That’s why we include Figure 4: it shows what actually happens in this scenario. If the number of particles is far lower than the number of minima, the particles in D(R)LE  are not enough to be a good approximation to a density of this type of space. Instead, they drift to the closest minima (exactly like they would in DEs!) and get stuck there for a long time. To show this phenomenon even more clearly, we have attached the original picture in Figure 3 together with a variant of the same experiment that uses only four particles to approximate the D(R)LE measures to this review. As is clear from that picture, when the number of particles is small relative to the number of minima, a naive interpretation of the theory is misguided; and D(R)LE behaves very similar to DEs. This is consistent with how the underlying equations evolve: When we discretise the evolution equations, what we get are basically slightly modified and randomized gradient descent schemes (see e.g. eq (7)). Theory (and experience from Langevin sampling for Bayesian posteriors) shows that as one keeps evolving the particle ensemble long enough, they will eventually escape and behave sufficiently differently from DEs—but in deep learning, we will not have the resources to keep the processes running forever. The naive expectation that we would get the type of behavior we see in Figure 2 in neural networks is therefore voided—instead, we should expect that D(R)LEs will act very similarly to DEs.
> 3. Thank you for pointing this out; we will improve the paper and make these points clearer. Specifically we will explain that all integrals are over $R^J$ and all gradients are with respect to $\theta$.
>
> #### Questions
> 1. The number of ensemble members is $N_E$ (e.g. $N_E=10$). This means we train 10 neural networks where each neural network has its own set of parameters. The index $n$ therefore would run from 1 to 10 and denote the set of parameters corresponding to the n-th neural network.
> 2. Yes, indeed. In Section 4.2 and Theorem 2 we show that under certain conditions our intuitions are validated and we obtain convergence results.
> 3. In Figure 2 we illustrate that the samples we obtain from implementing our method for DLE and D(R)LE are close to the optimal measures.
> 4. Yes, this is indeed the same.
> 5. See answer to Weakness 1.
> 6. Yes. Commonly, ‘non-parametric’ methods refer to the setting where the number of parameters can be arbitrarily high, and where in the limit of infinitely many parameters, one can recover an (infinite-dimensional) truth. In the context of our paper, the non-parametric object is indeed the targeted distribution itself, while the number of particles (of DE or D(R)LE) can be seen as the (arbitrarily large) number of parameters. As our theory shows, if the number of particles/parameters goes to infinity, we can recover the infinite-dimensional object we are targeting. This is different from parametric/parameterised problems, where there is not a natural way of making the parameter space arbitrarily large. E.g., typical variational families such as the family of normal distributions is parametric—but the collection of normal mixture distributions (with arbitrarily many and potentially infinitely many mixture components) would be non-parametric.
> 7. See Weakness 1.
> 8. See Weakness 2.
>
> #### Limitations
> See global response point in 4.

---

### Official Review · Reviewer_NdiF · 2023-07-06

**Soundness:** 3 good
**Presentation:** 2 fair
**Contribution:** 3 good
**Rating:** 8
**Confidence:** 3

**Summary:**

To improve the accuracy of the uncertainty quantification, the authors aim to provide a mathematically rigorous link between Bayesian inference, Variational Bayes methods and ensemble methods. In this work, methods s.a. variational inference, Langevin sampling and deep ensembles can be seen as particular cases of an infinite-dimensional regularised optimization problem formulated via Wasserstein gradient flows. They also provide a novel inference algorithm based on MMD and gradient descent in infinite dimensions plus regularisation.

The procedure takes place by reframing the usual finite-dimensional loss function problem into an infinite-dimensional one. This is done rewriting the original optimization problem using an infinite-dimensional problem over the set of probability measures $\mathcal{P}(R^J)$ and introducing a strictly convex regulariser to induce a unique global solution. This solution is assumed to not be too different from the solution to the original problem, which is controlled by a reference measure $P$. This leads to an interpretation of many different inference setups as particular cases of the optimization problem proposed. Therefore, this approach results as a combination of the proposals in Knoblauch et al. (2022) and Ambrosio et al. (2005).



**Strengths:**

* Good idea, could be interesting to the community were it proven in some other contexts.

* The formulation is clear and elegant thanks to the gradient flows and the usage of Wasserstein space. The usage of the thremodynamical formulation of free energy is very attractive as well.


**Weaknesses:**

* Altough the proposal is interesting and elegant, I think the experimental part of the paper does not provide enough evidence of the benefits related to this framework change. Results such as those present in Figure 3 could, in principle, be rivaled by previous methods s.a. [1] and [4], neither of which are discussed here. The authors maybe could provide a stronger motivation in this regard, and maybe try to encompass these other methods inside their framework.

* Some literature relevant to the topic at hand seems to be missing from the discussion, or at least should be discussed more thoroughly:

  *  Regarding the definition of infinite-dimensional GVI methods, I consider that other methods based on samples are left out and should be considered, such as [1,2,3]. These works may seem specially relevant due to the interest in implicitly-defined target $Q^*$, and in particular those that make use of the function-space formulation s.a. [1] or [4].

  * I think finite-dimensional GVI methods are misrepresented as they can be much more expressive than the selection made in Section 2.2 may lead to believe. I consider that this point should be addressed, and the discussion must be readjusted accordingly in order to highlight the benefits of the proposed approach without relying on this fact. As examples of this matter, please see references [4,5,6].

* The writing can be generally improved, since the paper can be at times hard to follow. This is just a consequence of the amount of information provided, which is a positive point, although in sections 3 and 4 could be polished further.

* (minor) The presentation could be improved, for example, by convering images to formulas s.a. in Figure 1 or the layout on the final page.

* (minor) Since Wasserstein spaces are such a crucial point of this, I would suggest devoting a bit more time to explain the basics of the concept in the main text itself and not fully depend on the sources.

(_References included in the "**Limitations**" section_)

**Questions:**


* Can this framework be used to describe methods that obtain an a-posteriori approximation of the predictive distribution via Laplace approximation or similar methods? As examples, please see [7,8]

* Since now inference is conducted without the guarantees provided by the Bayesian method, how should the distributions obtained be interpreted or used?

* I think other possible interesting regularisation choices would be Renyi divergences and also any proper scoring rule, defined in [9], which may solve issues related to the MMD and KL divergence.

(_References included in the **"Limitations"** section_)

**Limitations:**


* Since the Bayesian framework is abandoned, I fear there are no guarantees about the properties for the distributions obtained in the same sense as with Bayesian inference. Although can be somewhat justified by results, a lot more work is needed in this regard in methods that rely on this extensions (which is a problem for this paper, although definitely not exclusive to it).

* The paper is centred on theoretical developments, and as such, the theoretical discussion and argumentation is really interesting. However, and although it is not the core of the paper, the experimental phase leaves a lot to be desired in terms of justifying why this formulation change is needed.

---
**References**:

[1] Rodrı́guez-Santana, S., Zaldivar, B., & Hernandez-Lobato, D. (2022, June). Function-space Inference with Sparse Implicit Processes. In International Conference on Machine Learning (pp. 18723-18740). PMLR.

[2] Mescheder, Lars, Sebastian Nowozin, and Andreas Geiger. "Adversarial variational bayes: Unifying variational autoencoders and generative adversarial networks." International Conference on Machine Learning. PMLR, 2017.

[3] Santana, S. R., & Hernández-Lobato, D. (2022). Adversarial α-divergence minimization for Bayesian approximate inference. Neurocomputing, 471, 260-274.

[4] Ma, C., Li, Y., and Hernández-Lobato, J. M. (2019). “Variational implicit processes”. In: International Conference on Machine Learning, pp. 4222–4233.

[5] Sun, S., Zhang, G., Shi, J., and Grosse, R. (2019). “Functional variational Bayesian neural networks”. In: International Conference on Learning Representations.

[6] Ma, C., & Hernández-Lobato, J. M. (2021). Functional variational inference based on stochastic process generators. Advances in Neural Information Processing Systems, 34, 21795-21807.

[7] Deng, Z., Zhou, F., & Zhu, J. (2022). Accelerated Linearized Laplace Approximation for Bayesian Deep Learning. Advances in Neural Information Processing Systems, 35, 2695-2708.

[8] Antorán, J., Janz, D., Allingham, J. U., Daxberger, E., Barbano, R. R., Nalisnick, E., & Hernández-Lobato, J. M. (2022, June). Adapting the linearised laplace model evidence for modern deep learning. In International Conference on Machine Learning (pp. 796-821). PMLR.

[9] Gneiting, T., & Raftery, A. E. (2007). Strictly proper scoring rules, prediction, and estimation. Journal of the American statistical Association, 102(477), 359-378.

---

> ### Author Rebuttal · Authors · 2023-08-08
>
> #### Weaknesses
>
> [This is meant as a response to the first 2 bullet points]
> We thank the reviewer for their helpful suggestions. We will include the suggested references and give a thorough discussion in the final version of the manuscript. It allows us to contrast infinite-dimensional gradient-flow methods with infinite-dimensional parameter-space methods. The functions space literature focuses on a formulation of the problem in infinite-dimensional function spaces—but still uses a finite-dimensional gradient flow based on parameterisations to implement its algorithms. In contrast, our method operates on a finite-dimensional parameter space, but implements an infinite-dimensional gradient flow. The function space view has the benefit that the functional loss  is often convex and consequently the target $Q^*$ can be unimodal. However, the variational stochastic process still requires parameterization to be computationally feasible—and in this sense, function space methods are FD-GVI approaches. The resulting objectives require a good approximation of the functional KL-divergence (which is challenging), and lead to a typically highly non-convex variational optimization problem in the parameterised space. While good initialization strategies may well be able to overcome some of these issues, this type of tuning requirement is common amongst FD-GVI approaches, a direct result of non-convexity, and typically not grounded in theory. Moving away from relying on these types of tuning strategies (whose effect is often poorly understood) serves as another motivation to attempt an infinite-dimensional gradient flow procedure: in principle, it allows us to exploit the convexity in the space of probability measures directly. The derived inference algorithms and the resulting asymptotic guarantees give some evidence to this conjecture: they hold regardless of the chosen initialisation.
>
> That being said, we want to clarify that it is not our intention to question the validity of FD-GVI procedures or advocate for their abandonment: they are the dominant Bayesian deep learning paradigm for a reason, have considerable practical merits, and often work well in practice. We have mentioned and stressed this point more clearly in the new version of the manuscript. In spite of their practical utility, we do also believe that the mathematical challenges associated with FD-GVI serve as an additional motivation to investigate infinite-dimensional GD procedures. Indeed, doing this reproduced popular competing deep learning algorithms as diverse as DE and DLE—and even allowed us to derive DRLE, which provides a template for how our theory not only draws links btw existing deep learning algorithms, but can also inspire new ones.
>
> #### Questions
> 1. LLA relies heavily on a function space perspective, which we did not adopt in our work. It is possible that a thorough investigation of function space methods would lead to further connections but we consider this beyond the scope of this paper.
> 2. [This paragraph answers Question 2 and Limitation 1] Thank you for raising this point—this is indeed a very important point. We have included the following in the final version of the manuscript:
> > In essence, the core justification for these generalisations is that the very assumptions justifying application of Bayes’ Rule are violated in modern machine learning. In practical terms, this results in a view of Bayes’ posteriors as one—of many possible—measure-valued estimators $Q^∗$ of the form in (1). Once this vantage point is taken, it is not clear why one should be limited to using only one particular type of loss and regulariser for every possible problem. Seeking a parallel with optimisation on Euclidean domains, one may then compare the orthodox Bayesian view with the insistence on only using quadratic regularisation for any problem. While it is beyond the scope of this paper to cover these arguments in depth, we refer the interested reader to Knoblauch et al. (2022).
>
> 3. Again, the reviewer is pointing out a very interesting avenue for future research. In principle, it is possible to use other divergences such as the ones mentioned by the reviewer (and indeed all f-divergences). However, often the implementation of the gradient flow then requires us to have access to the evolved pdf of the samples at time $t$. This can generically be replaced with a kernel density estimator based on the samples available at time $t$. However, it is well-known that kernel density estimators suffer greatly from the curse of dimensionality. Therefore, it is practically infeasible to use them in the context of deep learning (as the parameter space over which we need to compute said kernel density estimators) is huge.
>
> #### Limitations:
> 1. See Question 2.
> 2. We agree that the paper does not provide a full analysis of empirical performances. One reason for this is the limited amount of space. That being said, the more important reason is that we do not intend the paper to be a thorough study of any particular ‘market-ready’ methodology that can compete with the fine-tuned algorithms prevalent in industrial scale deep learning. Instead, our focus is on theoretical insight into what existing methods for uncertainty quantification in deep learning are actually doing—and this is reflected by our emphasis on connecting Bayesian and non-Bayesian methods through the analytical lens the paper proposes. As part of this, we conducted a range of experiments in the experimental section (incl. on UCI regression tasks) to showcase the limitations of naively adopting the current framework for designing new algorithms. In this sense, we believe that our empirical investigation serves the main purposes of the paper: to highlight how the derived theory aligns with reality, and to explain any divergence between theory and reality to help follow-up research with exploiting our ideas in order to develop more efficient and more effective algorithms.

---

> > ### Comment · Reviewer_NdiF · 2023-08-12
> > **Brief response to the rebuttal**
> >
> > I want to thank the authors for their insightful responses and detailed comments on the reviews, including mine.
> >
> > After reading the rebuttals and going over parts of the article again, I'm really happy with what's been presented. I'm now even more convinced that we should accept this submission and I will update my review to reflect this. I consider this submission to be an interesting piece of work with important implications for future research.
> >
> > Thanks again for the good work!

---

### Official Review · Reviewer_6hMj · 2023-07-08

**Soundness:** 3 good
**Presentation:** 3 good
**Contribution:** 2 fair
**Rating:** 7
**Confidence:** 4

**Summary:**

The paper offers a viewpoint on deep ensembles as a (unregularized) Wasserstein gradient flow in the space of probability measures. This viewpoint enables new algorithms for deep ensembles (Langevin and repulsive via MMD), which are evaluated on some small datasets.

**Strengths:**

1) The paper is technically sound and well-written. Overall it was easy to follow.
2) While many similar ideas have been floating around in the literature, the precise presented view on deep ensembles seems novel, and I found Theorem 1 to be interesting.
3) While experiments on larger neural networks are missing, the effect of the proposed algorithms is clearly demonstrated in some controlled experiments and small data sets.

**Weaknesses:**

1) Perhaps the main weakness of the paper is the lack of a comparison of the new methods on large neural networks.

2) Many of the introduced tools (convexification via probabilistic lifting, Bayes with general divergence function, Wasserstein flows, etc.) are well-known.  But I believe Theorem 1 and Theorem 2 offer some new insights (in case they are really correct, see Questions).

**Questions:**

1) In section 2, it is written that one lifts the problem to a more "challenging space" -- but I would instead say that this space is much simpler.  The problem suddenly has a closed-form solution (Gibbs measure) that can be written down, one has convexity, etc.

2) The claim that the "infinite-dimensional regularised optimisation problem over the space of probability measures first introduced in Knoblauch et al. (2022)" seems rather quite bold -- the optimization problem (1) is a very fundamental one -- as discussed later in the paper (Section 2.1) there are many references. So perhaps this sentence in the introduction should be reshaped a bit?

3) In Theorem 1, is it really local minima or could it also be saddle-points? Couldn't a gradient flow in a nonconvex objective also get stuck at a saddle-point or local-maximum (when initialized at the maximum)? Imagine a landscape where we have a very large flat local maximum which has non-zero measure under that initial distribution.  Is this somehow excluded in the assumptions?

**Limitations:**

All limitations are addressed.

---

> ### Author Rebuttal · Authors · 2023-08-08
>
> #### Weaknesses
>
> 1. We understand the reviewer’s concern and agree that the paper does not provide a full analysis of empirical performances. One reason for this is the limited amount of space that 9 pages allow in order to comprehensively present our firmly grounded theoretical framework and a number of its technical aspects. That being said, the more important reason is that we do not intend the paper to be a thorough study of any particular ‘market-ready’ methodology that can compete with the fine-tuned algorithms prevalent in industrial scale deep learning. Instead, our focus is on theoretical insight into what existing methods for uncertainty quantification in deep learning are actually doing—and this is reflected by our emphasis on connecting Bayesian and non-Bayesian methods through the analytical lens the paper proposes. As part of this, we conducted a range of experiments in the experimental section (including on UCI regression tasks) to showcase the limitations of naively adopting the current framework for designing new algorithms. In this sense, we believe that our empirical investigation serves the main purposes of the paper: to highlight how the derived theory aligns with reality, and to explain any divergence between theory and reality to help follow-up research with exploiting our ideas in order to develop more efficient and more effective algorithms.
>
> 2.  We thank the reviewer for their comment. Indeed, we do not want to exaggerate our proposal’s novelty. While we agree with the reviewer that many of our contribution’s building blocks have been floating around in the literature, we are unaware of any work that has combined them in the way the current manuscript has. Bayesian (and generalized Bayesian) procedures typically focus on justifications that are centered around updating prior to posterior knowledge. Our focus is different: we instead focus on regularization in the space of probability measures, and on algorithms that allow us to solve the resulting problems. On a technical level, we also innovate by introducing WGFs for solving these types of problems. To the best of our knowledge, WGF has not previously been used to study how different regularisers translate into different properties of the inference algorithms; and how this in turn places inference algorithms as different as deep ensembles and variational Bayesian methods under the same overarching framework
>
>
> #### Questions
> 1. Thank you for pointing this out; the sentence is indeed misleading in a certain sense: The initial space here is the Euclidean space and the ‘more challenging’ space refers to the space of all probability measures on the Euclidean space. From a point grounded in analysis, the latter is indeed more challenging as it provides less structure: Unlike Euclidean spaces, it is infinite-dimensional, non-linear and does not have an inner-product structure, which makes a thorough analysis more challenging. And yet, the reviewer is also correct to point out that the optimization problem itself is much simpler thanks to its convexity on this (more challenging-to-analyse) space. We intended to describe precisely this trade-off: We trade the simple Euclidean space with the more complicated space of probability measures in order to obtain an ‘easier’ convex objective function. We will slightly rephrase this sentence in the new version of the manuscript to stress this point more clearly.
>
> 2. Thank you very much for making us aware of this—you are absolutely correct: this sentence should not have been in the final manuscript. We will ensure that this part of the paper is fixed by providing the adequate context. If you believe there are additional relevant papers related to these types of problems that we should be citing but that are currently not contained in the manuscript, we would be very appreciative of you letting us know so that we can include them.
>
> 3. This is an excellent question. Two assumptions prevent this behavior: First, we assume that every saddle point has at least one strictly negative eigenvalue. Second, we assume that the Lojasiewicz inequality is satisfied (Lemma 3 in Appendix). Intuitively, the first assumption guarantees that locally around a saddle-point, the domain of attraction has Lesbegue measure zero. Hence, if our initialisation measure $Q_0$ has a Lesbegue density, we will almost surely not be attracted to the saddle point and leave its domain of attraction. The Lojasiewicz condition then guarantees that once we are in the domain of attraction of a local minimum, we will stay there and converge eventually to it. For more details see [1]. Regarding the reviewer’s specific scenario: A loss with a flat local maximum that has non-zero measure under the initialization would violate the Lojasiewicz inequality—as the reviewer correctly hypothesizes, this setting is therefore not covered by the theory.
>
>
> [1] Lee, J. D., Simchowitz, M., Jordan, M. I., and Recht, B. (2016). Gradient descent only converges to minimizers. In Conference on learning theory, pages 1246–1257. PMLR.

---

> > ### Comment · Reviewer_6hMj · 2023-08-17
> >
> > Thanks for all the detailed clarifications -- I understand now much better Theorem 1 and its assumptions.

---

### Author Rebuttal · Authors · 2023-08-08

### General Response:

We want to thank all the reviewers for taking the time to read our manuscript so carefully and for providing valuable feedback that we believe will significantly improve the manuscript further. Overall, we have obtained a median score of 8, which is a wonderful reward for the countless hours that we have spent on this project and its technical content in the past year.
We summarize the reviewer’s main feedback below, and explain how we have addressed the points in the uploaded and updated version of the manuscript which implements some of the most called-for changes.

1. **Reviewer NdiF** has expressed a wish for a more thorough discussion of FD-GVI in our manuscript, and has raised concerns that our representation of parameterised variational methods may fall short; and **Reviewer 4EE1** raised a similar concern. We thank both reviewers for pointing out this oversight, and have added to the discussion on related literature in Section 2.2, which now includes a paragraph on the state-of-the-art on function-space inference, implicit variational inference strategies, and normalizing flows. We believe this discussion will further highlight that the manuscript’s intent is not to imply that FD-GVI methods are impractical; and that rather, its intent is to thoroughly highlight their conceptual shortcomings relative to ID-GVI methods (which are the main subject of our investigations).
**Reviewer 6hMj** has rightfully pointed out that a sentence regarding the origin of the studied optimization problem in the introduction needs some reshaping to avoid overstating the contribution in Knoblauch et al. (2022). We agree with the reviewer, and have adjusted this sentence.

2. **Reviewer NdiF** has raised some valid points regarding the interpretability and meaning of the non-Bayesian nature of the measure-valued estimators which we derive. We agree that such a discussion is useful and necessary—we have added it to Section 2.1.

3. Both **Reviewer 9edE and 4EE1** have asked questions regarding a quantitative analysis of the approximation error caused by finite time, finite samples and the use of unbiased estimators. We thank the reviewers for raising this, and have added the following discussion in Section 4.3:
> A notable shortcoming of Theorem 2 is its asymptotic nature. A more refined analysis would quantify how fast the convergence happens in terms of $N_E$, $T$, the SDE's discretisation error, and potentially even the use of unbiased estimators for the loss based on sub-sampling.
While the existing literature could be adapted to derive the speed of convergence for DRLE in $T$ (Ambrosio et al.,
2005, Section 11.2), this would require a strong convexity assumption on the potential $V$, which will not be satisfied for any applications in deep learning.
This is perhaps unsurprising: even for the Langevin algorithm—probably the most thoroughly analysed algorithm in this literature—no convergence rates have been derived that are applicable to the highly multi-modal target measures encountered in Bayesian deep learning  (Wibisono, 2019; Chewi et al., 2022).

4. **Several reviewers (6hMj, NdiF, AtWA)** pointed out that relative to the rest of the manuscript, its experimental section is comparatively limited. We agree that the paper does not provide a full comparison of empirical performances between different deep learning methods. The most important reason for this is that we do not intend the paper to be a thorough study of any particular ‘market-ready’ methodology that can compete with the fine-tuned algorithms prevalent in industrial scale deep learning. Instead, our focus is on theoretical insight into what existing methods for uncertainty quantification in deep learning are actually already doing. This is reflected by the title and by our emphasis on connecting Bayesian and non-Bayesian methods through the analytical lens the paper proposes. So while we did conduct a range of experiments in the experimental section (including on UCI regression tasks), this was done to showcase the usefulness of our framework as well as the limitations of naively adopting it for designing new algorithms—rather than as evidence that the proposed framework can outcompete prevalent deep learning paradigms. In this sense, we believe that our empirical investigation serves the main purposes of the paper: to highlight how the derived theory aligns with reality, and to explain any divergence between theory and reality to help follow-up research with exploiting our ideas in order to develop more efficient and more effective algorithms.

We have also included a number of smaller changes listed below:

1. Adding to why the MMD can encode useful properties as a regulariser as contrasted with the KLD (included in 4.3 already; **Reviewer 9edE**).

2. Being more explicit about what we mean by ‘challenging space’ (included in 2 already;  **Reviewer 6hMj**).

3. Explaining that all integrals are over the parameter space $R^J$ of $\theta$, and that similarly, all gradients are with respect to $\theta$ (included just before 2.1;  **Reviewer AtWA**)

4. Inclusion of a further setting for one of the experiments (the one for Figure 3) to make the messaging around the number of particles
relative to the number of local minima clearer & more explicit (**Reviewer AtWA**; the resulting experiments are can be found in the attached PDF).

In addition to changes that are already implemented, we will include further adjustments in the camera-ready version as listed below:

1. Adjustments to make the experimental section more readable and to state the purpose of the experimental section more clearly; with scope depending on space available (**Reviewer 9edE**)
2. Including a definition of the Wasserstein distance (**Reviewer NdiF, AtWA**),
3.  Mentioning the two key assumptions required for Theorem 1. (**Reviewer 4EE1**)
4. More explicitly mention why it is hard to solve the PDE in (4) (**Reviewer 9edE**)

---

### Decision · Program_Chairs · 2023-09-21

**Decision:**

Accept (oral)

**Comment:**

There has been a prevailing question as to why seemingly naive deep ensembles have surprisingly good properties for uncertainty quantification.  This paper tackles that question using an analysis through the lens of Wasserstein gradient flow and optimization in the space of measures.  Several variations of ensembling are derived within this framework.

The reviewers were unanimously in favor of acceptance, saying the work indeed provides a fresh and informative perspective about the behavior of deep ensembles.  The most common criticism amongst reviewers was that the experiments were too small scale and a bit confusing---which I think is a minor flaw (but one that the authors should look to improve, of course, especially wrt clarity).  There were also questions about the appropriateness / usefulness of Theorem 2, since it is an asymptotic result.

Notes for camera-ready version: D’Angelo & Fortuin [2021] also discuss deep ensembles with a repulsive term, obtained through the lens of Stein variational gradient descent / gradient flow in both parameter and function space.  Thus I was surprised to see their work mentioned only once and in passing.  I believe this paper goes beyond their work with a more rigorous treatment; yet I think the draft would be much improved to add more context about the relationship to their work.  This discussion should also be included along with the other related work mentioned in point 1 of the general rebuttal.